# von Neumann Subfactors and Non-invertible Symmetries

Xingyang Yu[1] and Hao Y. Zhang [2]

[1]Physics Department, Robeson Hall, Virginia Tech, Blacksburg, VA 24061, USA

[2]Kavli Institute for the Physics and Mathematics of the Universe (WPI), University of Tokyo, Kashiwa, Chiba 277-8583, Japan

May 20, 2025

## Abstract

We use the language of von Neumann subfactors to investigate non-invertible symmetries in two dimensions. A fusion categorical symmetry $\mathcal{C}$, its module category $\mathcal{M}$, and a gauging labeled by an algebra object $\mathcal{A}$ are encoded in the bipartite principal graph of a subfactor. The dual principal graph captures the quantum symmetry $\mathcal{C}'$ obtained by gauging $\mathcal{A}$ in $\mathcal{C}$, as well as a reverse gauging back to $\mathcal{C}$. From a given subfactor $N \subset M$, we derive a quiver diagram that encodes the representations of the associated non-invertible symmetry. We show how this framework provides necessary conditions for admissible gaugings, enabling the construction of generalized orbifold groupoids. To illustrate this strategy, we present three examples: $\mathrm{Rep}(D_4)$ as a warm-up, the higher-multiplicity case $\mathrm{Rep}(A_4)$ with its associated generalized orbifold groupoid and triality symmetry, and $\mathrm{Rep}(A_5)$, where $A_5$ is the smallest non-solvable finite group. For applications to gapless systems, we embed these generalized gaugings as global manipulations on the conformal manifolds of $c = 1$ CFTs and uncover new self-dualities in the exceptional $SU(2)_1/A_5$ theory. For $\mathcal{C}$-symmetric TQFTs, we use the subfactor-derived quiver diagrams to characterize gapped phases, describe their vacuum structure, and classify the recently proposed particle-soliton degeneracies.

# 1 Introduction

Quantum field theory (QFT) has a well-developed algebraic approach, particularly through von Neumann algebras in algebraic quantum field theory (AQFT). This framework assigns algebras of local observables to spacetime regions, encoding deep structural properties of the theory (see, e.g, [Haa96, BF04, BDFY15, Ded22] for reviews). However, traditional algebraic approaches primarily focus on local fields, whereas modern developments in quantum field theory, in both high energy and condensed matter physics, highlight the crucial role of global symmetries, which are often implemented by extended topological operators rather than local ones [GKSW15].

In two-dimensional (2D) QFTs, global (0-form) symmetries manifest through topological line operators that obey nontrivial fusion relations. These symmetries are often non-invertible, meaning they do not form a group but instead satisfy fusion rules governed by fusion categories [ENO02, BT18, CLS$^+$19]. Understanding the algebraic structure of these noninvertible symmetries is essential for studying conformal field theories (CFTs) and topological quantum field theories (TQFTs), as they constrain correlation functions, duality structures, and emergent topological orders.

In this work, we use the von Neumann algebraic approach, rather than focusing on local operator algebras, to investigate the algebraic structure of extended topological operator algebras, which are central to understanding global symmetries. We apply subfactor theory to study noninvertible symmetries in 2D QFTs, where fusion categories and their module categories play a crucial role in describing the algebraic structure of topological operators. Subfactor theory, first introduced by Jones in the context of von Neumann algebras , [Jon83], has been instrumental in advancing our understanding of these symmetries (see, e.g., [GS12a, BKLR15]), particularly in rational conformal field theory (RCFT) (see, e.g., [EK93, KLM01]) and topological quantum field theory (TQFT) (see, e.g., [KPS07, Kod19]). Using subfactors, we analyze the gaugings and representations of non-invertible symmetries, providing a systematic approach to understanding their impact on quantum systems, in both gapless and gapped

phases.

For gapless phases described by CFTs, gauging a (non-invertible) symmetry gives rise to a topological manipulation, rather than a local deformation, on its conformal manifold. We apply the subfactor approach to study the gauging of non-invertible symmetries, particularly focusing on the generalized orbifold groupoid (Brauer-Picard groupoid) [ENOM09, GK21, DLWW24] and its action on the conformal manifold of $c = 1$ theories. These considerations are crucial for understanding how gaugings modify the global structure of 2D CFTs, demonstrating the importance of non-invertible symmetries that have been overlooked in the early years of 2D CFTs. The algebraic structure of these gaugings is encoded in principal and dual principal graphs of subfactors, which describe the fusion rules of topological operators and algebra objects associated with gaugings. Using subfactor techniques, we analyze concretely how the gauging procedure works for the exceptional $SU(2)_1/A_5$ CFT and the $c = 1$ CFT at the Kosterlitz-Thouless point.

For gapped phases described by TQFTs, we use the subfactor approach to study representations of non-invertible symmetries, which play a central role in characterizing the vacua and their excitations. The representations of these symmetries are described by module categories [Ost03], which have an elegant derivation from a subfactor approach. We will show that given a principal graph of a subfactor, one can construct an associated quiver diagram encoding how the modules are acted by the simple objects in the fusion category. Physically speaking, they encode how the vacua transform under non-invertible symmetries, and how topological and local excitations combine as multiplets under non-invertible symmetry actions. A key consequence of this algebraic structure is the complete classification of particle-soliton degeneracies [CGSH24a, CHO24, CGSH24b] (see also [LMC91, Zam90] for earlier results) in all gapped phases, for which only the fully SSB phase has been considered in the literature. Through subfactor methods, we provide a systematic framework for classifying these degeneracies, offering new insights into applying the algebraic structure to the physical significance in gapped systems.

To be clear, the idea that subfactor theories correspond to fusion categories and even more general tensor categories is already well-established in the mathematical literature. See, e.g., [GS12a, BKLR15, BR19] and references therein. Our interest in this paper is to explain and apply this algebraic approach, to explicit examples and illustrate how they work in physically engaging contexts. To that end, as we are not aware of similar presentations in the literature, we will concretely describe the realization of the structure of non-invertible symmetries in the subfactor language in order to provide precise computations. Hopefully, the resulting insights into non-invertible symmetries – such as gaugings and representation theories – will be helpful for introducing the subfactor approach to a broader community.

# 2 Subfactor Approach to Non-invertible Symmetries

In this section, we give a minimal background on von Neumann Algebras and subfactors that is necessary for our analysis. We hope to elaborate the following points:

- A von-Neumann algebra is a $C^*$ algebra that equals to the double commutant of itself. In particular, any von-Neumann factor can be decomposed into factors (von Neumann factors that has trivial centers). Factors admit a well-known type classification by Murray and von Neumann into type I, II, III, depending on existance of non-trivial finite projectors and minimal projectors. In particular, subfactor of type III arises in the context of quantum field theories.

- Mathematically, a **von-Neumann factor of type III** has a (subcategory of) $\mathcal{C} = \mathrm{End}_0(N)$ of endomorphisms of $N$ forming a fusion category. In particular, an inclusion of $N$ into another type III factor $N, M$, known as **subfactors**, contains information about a Frobenius algebra object $A$ in $\mathcal{C}$ specifying its module category $\mathcal{C}_A$ and bimodule category $_A\mathcal{C}_A$.

- We use subfactors as a convenient and instructive mathematical tool to analyze a 2D CFT with fusion categorical symmetry $\mathcal{C}$. In this way, a subfactor $N \subset M$ encodes information about a gauging obtained by specified by condensing a line $A$ in half of the spacetime. This way, the possible interfaces and the quantum symmetry in the dual theory can both be understood via the subfactor approach.

- Independently, we review the more conventional interpretation of von-Neumann algebras and subfactors in algebraic QFT (AQFT), namely to describe algebra of local observables in a subregion of the physical spacetime. Under suitable physical assumptions (Haag duality to be specific), this algebra is given by a von Neumann algebra that is a factor of type III. In particular, a local symmetry can be described by the extension of local operators $N := \mathcal{A}(O) \subset M := \mathcal{B}(O)$.

- It is highly indicative that there should be fundamental physical connection between these two physical identification of subfactors (gauging generalized symmetries vs extension of local net of observables), which we hope to come back in the near future. However, for our current purpose, it is sufficient to be agnostic about such physical connection, and only to focus on the formal / mathematical correspondence.

## 2.1 von Neumann Algebra

A von Neumann algebra $A$ is a special type of $C^*$ algebra consisting of (bounded) operators $B(\mathcal{H})$ acting on a Hilbert space $\mathcal{H}$, defined over complex field and equipped a complex conjugation. To describe the extra structure that appear, we need the notion of the commutant, which is the full set

of bounded operators on the same Hilbert space that commute with the original algebra: $A' = \{T \in B(\mathcal{H}) | TS = ST, \forall S \in B(\mathcal{H})\}$.

A von Neumann algebra $A$ is a $C^*$ algebra (complex-valued algebra with conjugation) which is the double commutant of itself $A = A''$. We can further define its center as the intersection of itself with its commutant $\mathcal{Z}(A) = A \cap A'$. A von Neumann algebra with trivial center is called a *factor*.

Factors play a key role in the study of von Neumann algebra, since any von Neumann algebra can be decomposed a into factors in a unique way. Factors are classified into type I, II, and III originally carried out by Murray and von-Neumann [MvN36, MvN37, vN40, MvN43] (See [Sor24] for a physicist-friendly review). We can understand such a classification by the property of projection operators

- Type I factors are those with a minimal projector. They arise in quantum mechanical systems, where a minimal projector projects the system into a pure state.

- Type II factors are those with no minimal projectors but with finite projectors. Some type II factors arise in the study of quantum gravity, as pioneered by [LL23, CLPW23].

- Type III factors are those with neither finite nor minimal projectors. They arise in the study of local quantum field theories.

**Algebraic approach of QFT.** Even though we do not aim at making a physical connection between the algebraic approach of QFT versus generalized global symmetries, we still briefly summarize the interpretation of von Neumann subfactor under the algebraic approach of QFT in order to stimulate interest for future research along this line.

In the algebraic approach of QFT, we consider a fixed spacetime region $\mathcal{O}$, where the algebra of observables is denoted as $\mathcal{A}(O)$. In QFTs with Lagrangian descriptions, elements of $\mathcal{A}(O)$ are usually not the field itself, but an weighted average of field in a finite region. A minimal assumption is that such algebra gets larger when we enlarge the spacetime region (*isotonous* in mathematical terms), i.e., $\mathcal{A}(O_1) \subset \mathcal{A}(O_2)$ for $O_1 \subset O_2$. Such an algebra $\mathcal{A}(O)$ of local observable act on Hilbert spaces, which are in term representations $\pi(\mathcal{A}(O))$ of the algebra of observables by definition. We assume the existence of the unique vacuum representation $\pi_0$, which in particular satisfy Haag duality: [1]

$$\pi_0(\mathcal{A}(O)) = \pi_0(\mathcal{A}(O)')'. \tag{1}$$

---

[1]See [SSS25] for a recent discussion on the subtlety of these two conditions for theories with generalized global symmetries.

## 2.2 Subfactors

We consider subfactors $N \subset M$ of $N, M$ which are both von Neumann factors of type III. In this part, we mostly follow the presentation of [BKLR15][2]. In the end, the category of DHR homomorphisms will be interpreted as the categorical symmetries and their gaugings discussed in [BT18].

**Q-system.** The subfactors $N \subset M$ is characterized by a Q-system (a symmetric special Frobenius algebra in the fusion category $\mathcal{C}$, latter we put fusion category terminology in parenthesis), which is a triple

$$\mathbf{A} = (\theta, w, x) \tag{2}$$

where $\theta$ is an unital endomorphism of $N$ (object of the category $\mathcal{C}$), $w \in \mathrm{Hom}(\mathrm{id}, \theta), x \in \mathrm{Hom}(\theta, \theta^2)$ are a pair of intertwiners [3] (morphisms between corresponding object in $\mathcal{C}$) satisfying

$$\text{unit property}: \ 1_\theta = (w^* \times 1_\theta) \circ x = (1_\theta \times w^*) \circ x \tag{3}$$

$$\text{associativity}: \ (x \times 1_\theta) \circ x = (1_\theta \times x) \circ x \tag{4}$$

$$\text{Frobenius property}: \ xx^* = (1_\theta \times x^*) \circ (x \times 1_\theta) = (x^* \times 1_\theta) \circ (1_\theta \times x). \tag{5}$$

For a pair of factors $N \subset M$, we can consider homomorphism between them. In particular, the identity homomorphism will play an important role:

$$\iota : N \to M \tag{6}$$

which sends $n \in N$ to its image $\iota(n) \in M$ (which we denote as $n$ when no confusion arises).

For them we can define a dimension function that is additive under direct sum and multiplicative under compositions. The dimension $\dim(\iota) = \sqrt{[M : N]}$ by definition, where $[M : N]$ is the index of the subfactors $N \subset M$.

In fact, the introduction of subfactors can be motivated and explained in a highly diagrammatic fashion, which could make our explanation much more intuitive. The idea is to take the the above condition of a symmetric Frobenius algebra, which has conditions imposed by diagrams (3.1.1) - (3.1.3) of [BKLR15].

Now, the subfactors can be thought of as a general mathematical construction for such gaugeable algebra object. The idea is to resolve each line into a dark-shaded region, such that $N$ labels the original vacuum, while $M$ labels the new vacuum. (For the moment we are using this vague description on

---

[2]see [Bis16, BDVG22, BDVG23, Gio23] for more discussion on conformal nets and VOAs from the Algebraic QFT community.

[3]An intertwiner between two automorphisms $\alpha, \beta \in \mathrm{End}(N)$ is an element $t \in N$ such that $t \cdot \alpha(n) = \beta(n) \cdot t$ for any $n \in N$.

purpose, to make the discussion purely formal / mathematical.) All three ingredients in the subfactor $\mathbf{A} = (\theta, w, x)$ the following purposes:

- $\theta = \iota\bar{\iota}$ allows us to resolve one vertical line to a dark region, with left boundary encoding $\iota : N \to M$ and the right boundary encoding $\bar{\iota} : M \to N$.

- $w \in \mathrm{Hom}(Id, \theta)$ allows a strip of new vacuum to be created starting from a lower cap. Then $w^*$ allows a strip of new vacuum to terminate at an upper cap.

- $x = \in \mathrm{Hom}(\theta, \theta^2)$ allows us to split the strip new vacuum into two strips.

Then, the unit property, associativity, and Frobenius property thus guarantees that we can freely perform continuous deformations on the (boundary of) the new region. (See figure on page 21 of [BKLR15])

Before proceeding, we remark that in [BR19], a hypergroup (generalized version of fusion algebra by allowing non-integer coefficients) is defined for each pair of type III subfactors $N \subset M$ with finite index. This hypergroup always contains the fusion algebra of $M - M$ bimodules, but it contains some extra ingredients beyond $M - M$ bimodules under some conditions. [4]

**Module of a Q-system.** A module of a Q-system $\mathbf{A} = (\theta, w, x)$ (corresponding to elements in the category of right $A$-modules $\mathcal{C}_A$) is a pair $\mathbf{m} = (\beta, m)$ where $\beta$ is an object of the category $\mathcal{C}$, and $m \in \mathrm{Hom}(\beta, \theta\beta)$ an intertwiner, satisfying

$$\text{unit property} :(w^* \times 1_\beta) \circ m = 1_\beta \tag{7}$$

$$\text{representation property} :(1_\theta \times m) \circ m = x \times (1_\beta \circ m) \tag{8}$$

Furthermore, if $m^*m = 1_\beta$, then this module is called a standard module, which is automatic for irreducible modules. Two such modules $(\beta, m)$ and $(\beta', m')$ are equivalent if and only if there is an invertible $n \in \mathrm{Hom}(\beta, \beta')$ such that $m' \circ n = (1_\theta \times n) \circ m$. Such an equivalence is called a unitary equivalence if $n$ is unitary.

It was shown that, every standard model $\mathbf{m} = (\beta, m)$ of a simple Q-system $\mathbf{A} = (\theta, w, x)$ is unitarily equivalent to a standard module of the form $(\bar{\iota}\varphi, x)$, where $\phi : N \to M$ is a homomorphism.

Similarly, one can consider $\mathbf{A}_1 - \mathbf{A}_2$ bimodule of two Q-systems $\mathbf{A}_1 = (\theta_1, w_1, x_1)$, $\mathbf{A}_2 = (\theta_2, w_2, x_2)$ as a triplet $(\beta, m_1, m_2)$ where $m_1 \in \mathrm{Hom}(\beta, \theta_1\beta)$ and $m_2 \in \mathrm{Hom}(\beta, \beta\theta_2)$, such that $(\beta, m_1)$ is a left $\mathbf{A}_1$-module and $(\beta, m_2)$ is a right $\mathbf{A}_2$-module. It was proven by [EP12] that every standard bimodule is unitarily isomorphic to a bimodule of the form $\beta = \bar{\iota}_1\varphi\iota_2$, where $\varphi : M_2 \to M_1$ is a sub-homomorphism (obtained via projection) of $\iota_1\rho\bar{\iota}_2$ for some $\rho$.

---

[4]We thank Yuji Tachikawa for pointing us to this paper. See also [KTZ24] which analyzed non-invertible symmetries using hypergroups.

In the special case of $\mathbf{A}_1 = \mathbf{A}_2 = \mathbf{A}$, we get $\mathbf{A} - \mathbf{A}$ bimodules, which corresponds to elements in the dual fusion category $_A\mathcal{C}_A$ after the gauging. Instead, if we take $\mathbf{A}_2$ to be trivial, then we recover the case of module category discussed above.

| Subfactor $N \subset M$ | $(\mathcal{C}, A)$ | Physics theory $\mathcal{T}$ | principal graph |
|---|---|---|---|
| $N - N$ bimodules | $\mathcal{C}$ objects | topological lines in $\mathcal{T}$ | even part |
| $M - M$ bimodules | $_A\mathcal{C}_A$ objects | topological lines in gauged theory $\mathcal{T}/A$ | dual even part |
| $N - M$ bimodules | $\mathcal{C}_A$ objects | interfaces from gauging; right boundaries | odd part |
| $M - N$ bimodules | $_A\mathcal{C}$ objects | interfaces from dual gauging; left boundaries | (dual) odd part |

Table 1: Physical objects of topological lines, gauging interfaces and boundary conditions described both in the subfactor language and in the fusion category language.

We summarize the above content of modules and bimodules of subfactors, schematically, in Table 1.

**Computing the Principal Graphs.** We now explain how we compute the principal graph of the subfactor $N \subset M$ which is an important tool that is necessary in the analysis. In particular, in [GS12a], principal graphs played a key role for them to identify a fusion category with same fusion rule as the Haagerup fusion category, but is different from the latter.

The fusion matrix $F$ for an object in the fusion category $\gamma \in \mathcal{C}$ is defined as $F_{ij}^\gamma = (\gamma\xi_i, \xi_j)$, which is a square matrix. On the other hand, the module fusion matrix for a module $\kappa$ is $A_{ik}^\kappa = (\kappa\xi_i, \eta_k)$, where $\eta_k$ runs over the full list of irreducible module categories. Here $A_{ik}^\kappa$ is not necessarily square, since the two indices $i, k$ run over different sets of objects. The index $i$ run over all $N - N$ bimodules, also known as even parts. The index $k$ instead run over all $N - M$ bimodules, also known as odd parts. For $\gamma = \theta$ the symmetric Frobenius algebra object $\theta = \iota\bar\iota$ coming from an module $\iota$, the fusion matrix is related to the module fusion matrix via a simple relation:

$$F^\theta = A^\iota (A^{\bar\iota})^T, \tag{9}$$

which can be shown as

$$F_{ij}^\theta = (\theta\xi_i, \xi_j) = (\bar\iota\iota\xi_i, \xi_j) = (\iota\xi_i, \iota\xi_j) = \sum_k (\iota\xi_i, \eta_k)(\eta_k, \iota\xi_j) = A_{ik}^\iota (A^{\bar\iota})_{kj}^T. \tag{10}$$

By taking $A^\iota$ as the adjacency matrix, we get a graph which we call the *principal graph* of a subfactor. Here the $N - N$ bimodules are called *even part* of this principal graph, and the $N - M$ bimodules are callad *odd parts* of this principal graph.

On the dual side, we also have a dual fusion matrix $F^{(D)}$, and a dual module fusion matrix $A^{(D)}$ relating the $M - M$ bimodules with the $M - N$ bimodules. The $M - N$ bimodules are in one to one correspondence with the $N - M$ bimodules, so we oftentimes refer to both of them as odd parts, without

explicitly using the word "dual". Diagrammatically, we can glue the adjacency graph (principal graph) of $A$ with that of the adjacency matrix (dual principal graph) of $A^{(D)}$ into a single diagram, see Figure (1) as an example where the principal graph and dual principal graph of Ising fusion category are glued together.

**Computing the Quiver Diagrams.** Given a subfactor principal graph encoding both a fusion category and its module category for certain algebra object $A$, we can obtain how modules transform under the fusion category objects, serving as representations for the fusion category.[5] For a given principal graph with $n$ even part vertices and $m$ odd part vertices, one can obtain $n$ quiver diagrams, each of which have $m$ quiver nodes. For the even part vertex labeled by $e_i$ $(i = 1, \cdots n)$, its action on odd part vertex $o_a$ $(a = 1, \cdots, m)$ read

$$e_i : o_a \to \sum_{b=1}^{m} c_b o_b. \tag{11}$$

The coefficient $c_b$ gives the number of arrows from $o_a$ to $o_b$. From the fusion categorical symmetry perspective, the coefficients $c_b$ are known to give rise to Non-negative Integer Matrix representations (NIM-reps) [FS03, CRSS23, DLWW24].

### 2.2.1 Physical Applications

Instead of giving a detailed general discussion of physically applications of subfactors here, we provide just a lighting overview and refer the reader to sections where we illustrate our idea via explicit examples.

- One application is building principal graphs as a necessary condition for a gaugeable algebra object. Given a non-simple object in a fusion category, we can compute its reduced fusion matrix and solve Eq. (10). If there does not exist a non-negative integer solution, then this object is not a gaugeable object. See Section 5.2.1, where we use this method to excluded certain objects as gaugeable ones.

- If a principal graph exists, one can extract the information of odd parts and perform further consistency checks. See Section 6.1, where we compare ratios of dimensions of odd parts to determine gaugeable algebra objects.

- After determining the algebra objects from subfactors, one can apply these gaugings as global manipulations on the conformal manifold of CFTs. Self-dualities under gauging lead to extended fusion-categorical symmetries. See Section 5.2.2 and 6.2.

---

[5]In many contexts, the terms module and representation are interchangeable, e.g. for a ring structure.

- The quiver diagrams from subfactor principal graphs have a nice physical interpretation as vacuum structure of the TQFTs with non-invertible symmetries [KWZ22, HLS21], as well as the particle-soliton degeneracies [CGSH24a] of the associated gapped phases. See Section 4, 5.1.2, 5.2.3 and 6.3.

## 2.3   Example: Ising fusion category

This content summarize concrete examples from [BKLR15] (their section 3).

The Ising fusion category is given by $[id], [\tau], [\sigma]$ that is the category of the DHR endomorphisms in the Ising CFT, satifying the fusion rules of $[\tau^2] = [id], [\tau \circ \sigma] = [\sigma \circ \tau] = [\sigma], [\sigma^2] = [id] \oplus [\tau]$.

The tensor category is specified by choosing one object in each equivalence class, and one intertwiner in each isomorphism class. Here, we choose $r \in \mathrm{Hom}(id, \sigma^2), r \in \mathrm{Hom}(\tau, \sigma^2)$ to be a pair of orthogonal isometries satisfying $rr^* + tt^* = 1$, and $u \in \mathrm{Hom}(\sigma, \sigma\tau) = \mathrm{Hom}(\sigma^2, \sigma^2)$ can be chosen to be $u = rr^* - tt^*$.

One can impose $\sigma^2(a) = rar^* + t\tau(a)t^*$, which fixed $\sigma$ up to a sign. It turns out that this sign has to do with F-symbol data: choosing $\sigma(r) = 2^{-1/2}(r + t), \sigma(t) = 2^{-1/2}(r - t)u$ gives the Ising fusion category, while choosing its opposite $\sigma(r) = -2^{-1/2}(r + t), \sigma(t) = -2^{-1/2}(r - t)u$ gives the fusion category associated with $su(2)_2$ current algebra. We hope to come back to a systematic study on this point in the future.

There are two Q-system in the Ising subfactor: $(\theta = id, w = 1, x = 1)$ implementing the trivial gauging, and $\theta = \sigma^2, w = 2^{1/4}r, x = 2^{1/4}\sigma(r) = 2^{-1/4}(r + t)$ implementing the non-trivial self-dual gauging. Here the normalization factors comes about since we want to define some projectors that sum up to identity. This extensionis controlled by $M = \iota(N) \wedge \psi$ and explanations, where $\psi = 2^{1/4}\iota(t^*v)$ is such that $\psi^2 = 1, \psi = \psi^*, \psi(\iota(n)) = \iota(\tau(n))\psi$.

For the non-trivial Q-system in the Ising factor, its bimodules can be obtained as follows:

- id-id bimodules correponds to elements in the ising category $id, \sigma, \tau \in \mathcal{C}$

- id-A bimodules (right $A$-modules) arising from $\varphi = \rho\bar{\iota}$. This include an identity right A-module with $\rho = id$ or $\rho = \tau$, and a reducible non-trivial right A-module characterized by $\varphi_\sigma(n) = \sigma^3 n, \ \varphi_\sigma(\psi) = rur^* - tut^*$. The latter module split into $\varphi_+ = r\varphi_\sigma r^*$ and $\varphi_- = t\varphi_\sigma t^*$ with projectors $r, t$, so that $\varphi_\pm(n) = \sigma n, \varphi_\pm(\psi) = \pm u$.

- A-A bimodules arising from $\varphi = \iota\rho\bar{\iota}$. Here $\rho = id, \tau$ again gives rise to equivalent bimodules, but this time it splits into $\phi_+ = id$ and $\phi_- = \sigma$ with projectors $\frac{1}{2}(1 \pm \psi)$. The third A-A bimodule with $\rho = \sigma$ would split into two bimodules $\varphi_\pm(n) = \sigma n, \varphi_\pm(\psi) = \pm u$ with projectors $r, t$, but this time, they are equivalent bimodules by the equivalence $\psi \in \mathrm{Hom}(\varphi_+, \varphi_-)$ (Compare to the

id-A bimodule case, we are now having different equivalence relations.) There three elements are elements of the dual Ising category $_A\mathcal{C}_A$.

The principal graph of this Ising category is given in Figure 1.

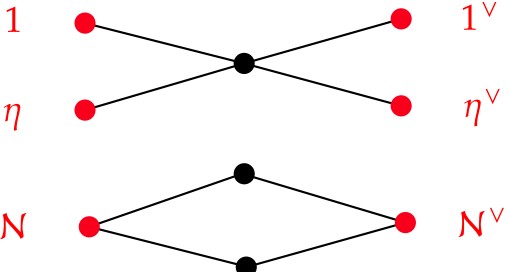

Figure 1: Principal diagram and dual Ppincipal diagram for the subfactor corresponding to gauging $1 + \eta$ in the Ising fusion category.

Given the bipartite property of the principal graph, we can compute the quiver diagram encoding how Ising category objects act on A-modules. This can be done by labeling each odd part vertex by even part vertices it connecting to,

$$o_1 : 1 + \eta, \ o_2 : \mathcal{N}, \ o_3 : \mathcal{N} \tag{12}$$

and then computing the fusion by each even part vertex, i.e., each simple object in the Ising category. The result reads[6]

$$
\begin{aligned}
&1 : o_a \rightarrow o_a, \ i = 1, 2, 3, \\
&\eta : o_1 \rightarrow o_1, \ o_2 \leftrightarrow o_3, \\
&\mathcal{N} : o_1 \rightarrow o_2 + o_3, \ o_2 \rightarrow o_1, \ o_3 \rightarrow o_1.
\end{aligned}
\tag{13}
$$

This gives rise to a representation of the Ising category, which can be summarized into quiver diagrams shown in Figure 2, one for each object in the Ising category. Each node corresponds to a $o_i$, while the links between them imply the fusion of the Ising category objects on them.

Furthermore, one can construct the (dual) principal graphs and the quiver representation for the other Q-system ($\theta = id, \omega = 1, x = 1$), associated with the trivial gauging. Although the principal graph look different from 1, the quiver diagrams are equivalent to Figure 2. This implies that these two

[6]There seems to be an ambiguity when acting $\eta$ on $o_2$ and $o_3$. In fact, a more precise way to label the even part connecting to $o_2$ and $o_3$ is using $\mathcal{N}$ and $\eta\mathcal{N}$, respectively. With the $\eta$ grading, the transformations of $o_2$ and $o_3$ under $\eta$ are determined.

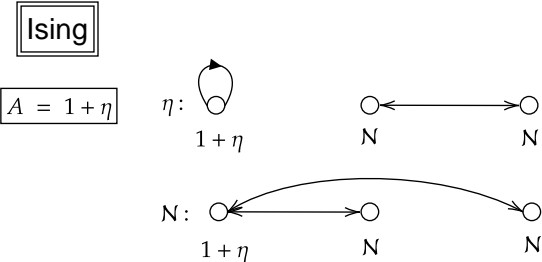

Figure 2: The quiver diagram of actions of elements in Ising fusion category on all its module categories. These three dots are the module categories for the Ising category under the algebra object $A = 1 + \eta$, which also appeared in Figure 1 and (12).

Q-systems have the same modules/representations, which by definition (see, e.g., [BKLR15]) means they are Morita-equivalent.

## 2.4 Example: $\mathrm{Vec}_G^\omega$

The fusion category obtained by a finite group $G$ (with anomaly $\omega \in H^3(G, U(1))$) is oftentimes denoted by $Vec_G^\omega$. Inside this fusion category, an algebra object $A(H, \psi)$ can be defined by a subgroup $H \subset G$ and a discrete torsion $\psi \in H^2(H, U(1))$ such that $\omega|_H$ is trivial. Our presentation follows that of [Ost06].

In this case, the $A(H_1, \psi_1) - A(H_2, \psi_2)$ bimodule are $H_1, H_2$ bimodule with a 2-cocycle given by

$$\psi^g(h, h') = \psi_1(h, h')\psi_2(g^{-1}h'^{-1}g, g^{-1}h^{-1}g)\omega(hh'g, g^{-1}h'^{-1}g, g^{-1}h^{-1}g)^{-1}\omega(h, h', g)\omega(h, h'g, g^{-1}h'^{-1}g)$$

(14)

More explicitly, each $A(H_1, \psi_1) - A(H_2, \psi_2)$ bimodule is specified by an $H_1 - H_2$ double coset inside G, together with a projective representation of $H^g = H_1 \cap gH_2g^{-1}$ (the centralizer of $g \in G$) with 2-cocycle $\psi^g$, where $g$ is any representative of this $H_1 - H_2$ double coset.

We would need to explicitly compute the fusion rule of such bimodules. Fortunately, we would only need to deal with the case where the anomaly $\omega$ is trivial. In this case, an explicit formula can be found in [KMY97].

## 3 Physical Applications

In this section, we will present a overview of applications of the subfactors approach to non-invertible symmetries to physically interesting scenarios.

## 3.1 Necessary conditions for gaugeable algebra objects

An immediate application underlies the following correspondence

$$\text{Q-system of a von Neumann algebra}$$
$$\longleftrightarrow \text{Symmetric special Frobenius algebra in the fusion category } \mathcal{C} \qquad (15)$$
$$\longleftrightarrow \text{Gauging of (non-)invertible } \mathcal{C} \text{ symmetry}$$

We already discussed the first half of the above correspondence in 2.2, while the second half can be found in [FRS02] (see also [CLS24, PLRS$^+$24a, DLWW24] for more recent explanation). The gauging of the $\mathcal{C}$ symmetry is specified by an algebra object $A = \sum_i c_i L_i$, which is a weighted sum of simple objects $L_i$ in $\mathcal{C}$. Given the physical interpretation of $L_i$ as simple topological line operators, physically speaking, gauging is to sum over the topological line configuration labeled by $A$. We thus conclude a necessary condition for $A$ being a gaugeable algebra object:

*If $A$ is a gaugeable algebra object of $\mathcal{C}$ symmetry, it must allows a principal graph. In other words, Eq.(9) mush has non-negative integer solution.*

See Section 5.2.1, where we use the above condition to exclude certain objects as gaugeable ones.

There are also cases that a complicated fusion categorical symmetry $\mathcal{C}$ is Morita-equivalent to a simple one, e.g, the invertible $\text{Vec}_G$ symmetry. If the gauging from $\text{Vec}_G$ to $\mathcal{C}$ is known, then the dual gauging from $\mathcal{C}$ to $\text{Vec}_G$ is highly constrained from the subfactor language. More generally, consider that gauging an algebra object $A$ of $\mathcal{C}$ has the dual quantum symmetry $\mathcal{C}'$, whose subfactor principal graph is known. Searching the dual gauging of $A'$ of $\mathcal{C}'$ translates into searching the dual principal graph for that subfactor. The key property of the dual principal graph is that it shares the same odd part (number and dimension) with the principal graph, underlying the fact that the odd part encodes not only modules, but also $N - M$ and $M - N$ bimodules. We thus have the following necessary condition for the unknown gaugeable algebra object $A'$ of $\mathcal{C}'$

*For a given gauging $A$ of $\mathcal{C}$ whose principal graph is $P$, its dual gauging $\mathcal{A}'$ of $\mathcal{C}'$ must has the principal graph $P'$, whose odd part has the number and dimensions as those of $P$.*

See Section 6.1, where we use the above method by comparing ratios of dimensions of odd parts to determine gaugeable algebra objects.

## 3.2 Global manipulations and self-dualities on conformal manifolds

One of the most important properties of a CFT is its conformal manifold. In addition to the local deformation along the manifold, there are also global manipulations realized by gauging[7]. For example,

---

[7]There can be interplays between local deformations and global manipulations on the conformal manifold. See, e.g.,[Lam25].

in 4D $\mathcal{N} = 4$ $SU(N)$ Super Yang-Mills theory, in addition to turning on marginal deformation with respect the complexified gauging coupling $\tau$, one can gauge its one-form $\mathbb{Z}_N$ symmetry to map the theory $SU(N)[\tau] \to PSU(N)[\tau] \cong SU(N)[-1/\tau]$ [AST13, KZZ22, LYZ24], which is a global manipulation from $\tau$ to $[-1/\tau]$.

In this work, we will focus on $c = 1$ conformal manifolds [Gin88], especially two of its special points. One is the Kosterlitz-Thouless point, which is the junction between the circle branch and the orbifold branch, while the other is the isolated point $SU(2)_1/A_5$, which is the orbifold theory of $SU(2)_1$ with the largest orbifold group. Using subfactor methods we introduced in the last subsection, we find out various gaugings of these two special points, and build the corresponding global manipulations on the $c = 1$ manifolds.

Gauging an algebra object $A$ inside a categorical symmetry $\mathcal{C}$ with a choice of discrete torsion / algebra structure will lead to a dual symmetry. This idea goes back to the notion of "quantum symmetry" first introduced by [Vaf89], where performing an orbifolding by a finite group $G$ will lead to a quantum symmetry $\mathrm{Rep}(G)$, which is not a group when $G$ is non-Abelian. Trying to describe $G$ and $\mathrm{Rep}(G)$ in the same mathematical framework leads to the notion of fusion categorical symmetry [BT18, CLS$^+$19]. Conceptually, the dual fusion category is understood as the category of $_A\mathcal{C}_A$ bimodules, and all possible gaugings are given by module categories $\mathcal{C}_A$ of right $A$-modules, corresponding to condensing a mesh of $A$-lines in half of the spacetime. However, the practical computation for such dual category and module category data is very challenging in general, and one needs other mathematical tools to even do this computation. Notably, in [GS12b], the authors determined all dual categories in Haagerup fusion category using the subfactor approach, where they found a fusion category ($\mathcal{H}_3$ in their notation) that has the same fusion rule but different F-symbols compared to the Haagerup fusion category. Motivated by their effort, we also use subfactor approach to determine the dual fusion categories of gaugings arise in the physical setup of $c = 1$ CFTs.

In addition, specific points of the conformal manifold may stay fixed under gauging. This situation is called a self-dual gauging of the CFT, with the primary example as the Kramer-Wannier duality defect in Ising CFT [FFRS04]. Thanks to the isomorphism between the original gauging and the dual gauging, we can perform such a gauging in half of the spacetime, therefore creating a topological interface between the two theories. When it is possible, the topological duality defect actually enhances the fusion categorical symmetry, also giving its fusion products with all existing topological lines. In $c = 1$ CFTs, there has already been suggestive results of such enhancement in dual local operator side in [DVVV89], which in modern languages means that a $\mathrm{Rep}(A_4)$ symmetry is enhanced to an $\mathrm{Rep}(SL(2,3))$ symmetry[8]. More recently, such an enhancement in generalized symmetries are explicitly

---

[8]Following the motivation of [DVVV89] by viewing the c = 1 CFT as a subsector on string worldsheet, it is interesting to look for spacetime interpretation of such symmetry following the line of [KTZ24, HMM$^+$24, PSY24, CPS$^+$25], and

observed in [PLRS+24b, CLS24, DLWW24, LSZ24, LSZ25]. We use the subfactor approach to extend this computation to the only $\mathfrak{e}_8$ (under ADE classification of orbifold models of $SU(2)_1$) exceptional point of $SU(2)_1/A_5$ in $c = 1$ CFTs associated to an unsolvable finite group $A_5$, where we find a self-dual gauging which extend the $\text{Rep}(A_5)$ symmetry, as we elaborate in Section 6.

## 3.3   Classification of gapped phases and particle-soliton degeneracies

In addition to CFTs, another physical context we are interested are TQFTs and gapped phases they describe[9]. In particular, we will focus on axiomatic TQFTs having non-invertible symmetries, describing gapped phases protected by non-invertible symmetries.

It is known that for a given $\mathcal{C}$, there is a one-to-one correspondence between its gaugeable algebra objects $A$ to $\mathcal{C}$-symmetric TQFTs [HLS21, KWZ22]. This can be understood from the SymTFT construction as follows [BBPSN25, PR24]: Gauging an algebra object $A$ amounts to changing the symmetry boundary (which is topological) from $\mathcal{L}$ to $\mathcal{L}'$, where $\mathcal{L}$ and $\mathcal{L}'$ are Lagrangian algebras for the SymTFT. Instead of specifying the Lagrangian algebra for only the symmetry boundary, one can put Lagrangian algebras for both symmetry and the physical boundary, which engineer a topological theory in 2D. Fixing the symmetry boundary $\mathcal{L}$ provides a certain fusion category $\mathcal{C}$, then all possible Lagrangian algebras on the physical boundary lead to a classification of the $\mathcal{C}$-symmetric TQFTs. We refer the reader to [HLS21, BBPSN25] for more details[10].

As discussed in Section 3.1, we are able to use the subfactor approach to search gaugeable algebra objects for a given $\mathcal{C}$. This thus allows us to provide a complete classification of the $\mathcal{C}$-symmetric TQFTs from the Q-systems, specified by the principal graphs.

A natural follow-up question is: what information can we extract from the Q-system of the subfactor for the corresponding gapped phase? To answer this question, recall that one of the defining data of a $\mathcal{C}$-symmetric TQFT is its boundary conditions. As we summarized in Table 1, mathematically speaking, these boundaries are given by the module categories. Utilizing the fact that the module category is concluded in the odd part of the subfactor principal graph, we have the following correspondence:

$$
\begin{aligned}
&\text{Odd part of principal graph} \\
\longleftrightarrow\ &\text{Module category of } \mathcal{C} \\
\longleftrightarrow\ &\text{Boundaries of } \mathcal{C} - \text{symmetric TQFTs}
\end{aligned}
\tag{16}
$$

In addition to boundary conditions for spatial-like direction, from a time-like direction perspective,

---

to compare it with analysis of topological operators in the string theory spacetime as studied in [HHTZ23, HHT+23, Yu24, Zha24].

[9]We will use "TQFTs" and "gapped phases" interchangeably in this work, without worrying about the exotic topological phases such as fractons.

[10]Correspondence between algebra objects and Lagrangian algebras for explicit examples can be found in [FY24, PR24]

these boundaries can be regarded as boundary states of the TQFT. Therefore, each simple object in the module category, i.e. a vertex in the odd part of the subfactor, corresponds to a vacuum state for the associated TQFT. The action of the $\mathcal{C}$ symmetry on these vacua, is then characterized by the quiver diagrams obtained from the principal graph. For example, the quiver diagrams in Figure 2 have the following physical interpretation: The Ising categorical TQFT describes a SSB phase with three vacua. The $\eta$ topological line maps one vacuum to itself, while switching the other two vacua. The $\mathcal{N}$ line maps one vacuum to the other two vacua and vice verse. The same quiver diagram alternatively describes how boundary conditions of the TQFT transforming under the Ising categorical symmetry[11].

These quiver diagrams labeling representations of $\mathcal{C}$ have a further physical interpretation: They classify the particle-soliton degeneracies. Briefly speaking, consider the TQFT quantized on the spatial manifold $\mathbb{R}$, then at $-\infty$ and $+\infty$ there can be two vacua states $|\Omega_-\rangle$ and $|\Omega_+\rangle$ assigned respectively[12]. Therefore, at a given time slice, we have the following correspondence

$$
\begin{aligned}
\text{particle} : |\Omega_-\rangle = |\Omega_+\rangle, \\
\text{soliton} : |\Omega_-\rangle \neq |\Omega_+\rangle.
\end{aligned}
\tag{17}
$$

This is due to the fact that particle is the local excitation for a given vacuum, while the soliton is the topological configuration transiting different vacua. We refer the reader to [CGSH24a, CHO24] (see also [CCK24, CCK25]) for a more detailed discussion on soliton representations under non-invertible symmetries.

Now come back to our subfactor approach. It is now straightforward to see that the quiver diagrams, derived from principal graphs, have exactly the same physical meaning as those in [CGSH24a]. Namely, given the nodes are module category objects labeling vacuum states of a gapped phase, the link for a single node is the particle, while the link connecting two nodes is the soliton! Remarkably, since we exhaust the TQFTs for a given $\mathcal{C}$ via searching for all its Q-systems, we provide a complete classification of particle-soliton degeneracies for $\mathcal{C}$-symmetric gapped phases, compared to the previous work in the literature where only the fully SSB phase is considered. Indeed, in this work we classify particle-soliton degeneracies for gapped phases with $\mathrm{Rep}(D_4)$ (Section 4), $\mathrm{Rep}(A_4)$ (Section 5.1.2), anomalous triality $\mathcal{C}^{\mathcal{T}}$ (Section 5.2.3), as well as $\mathrm{Rep}(A_5)$ (Section 6.3).

---

[11]For another example, see [BCPSN24] for the quiver representation of boundaries transforming under the $\mathrm{Rep}(S_3)$ symmetry.

[12]The category theory underlying this is the isomorphism between the representation of the strip algebra and the module category: $\mathrm{Rep}(\mathfrak{Strip}(\mathcal{C})) \cong \mathcal{M}$. We refer the reader to [CRZ24, CHO24] for more details on this point.

# 4 Rep($D_4$)

In this section, we apply the subfactor theory to investigate the Rep($D_4$) symmetry. While all possible gaugings and their associated module categories have been previously derived in the literature (e.g., [DLWW24, PLRS$^+$24b]), this example serves as a useful warm-up to illustrate the subfactor approach. In addition, we provide a complete classification of particle-soliton degeneracies in Rep($D_4$)-symmetric gapped phases, which, to our knowledge, has not appeared in the literature.

The dihedral group $D_4$ of order 8 is defined by

$$\langle a, b \mid a^4 = b^2 = (ab)^2 = 1 \rangle. \tag{18}$$

The fusion category of its regular representations, Rep($D_4$), consists of five simple objects: $1, \eta, \eta', \eta\eta'$, and $\mathcal{N}$, satisfying the abelian fusion rules

$$\eta \times \eta = \eta' \times \eta' = 1, \; \mathcal{N} \times \mathcal{N} = 1 + \eta + \eta' + \eta\eta', \; \mathcal{N} \times \eta = \mathcal{N} \times \eta' = \mathcal{N}, \tag{19}$$

which place it in the Tambara–Yamagami (TY) category of $\mathbb{Z}_2 \times \mathbb{Z}_2$ [TY98].

Since Rep($D_4$) arises as the quantum symmetry of gauging a $D_4$ symmetry, it is a group-theoretical fusion category. Its gaugings correspond one-to-one with the gaugings of $D_4$. As a group, the possible gaugings of $D_4$ are classified by its subgroups (up to conjugacy classes) and the choice of discrete torsions. In total, there are eleven distinct gaugings of Rep($D_4$) (see, e.g., [DLWW24]), associated with the following algebra objects $A$:

$$1, \tag{20}$$

$$1 + \eta, \; 1 + \eta', \; 1 + \eta\eta', \tag{21}$$

$$(1 + \eta + \eta' + \eta\eta')_2, \tag{22}$$

$$1 + \eta + \mathcal{N}, \; 1 + \eta' + \mathcal{N}, \; 1 + \eta\eta' + \mathcal{N}, \tag{23}$$

$$(1 + \eta + \eta' + \eta\eta' + 2\mathcal{N})_1, \; (1 + \eta + \eta' + \eta\eta' + 2\mathcal{N})_2, \; (1 + \eta + \eta' + \eta\eta' + 2\mathcal{N})_3. \tag{24}$$

Here, the subscript "2" in $(1 + \eta + \eta' + \eta\eta')_2$ denotes a discrete torsion (i.e., the non-trivial element in $H^2(\mathbb{Z}_2 \times \mathbb{Z}_2, U(1)) = \mathbb{Z}_2$) distinguishing it from $(1 + \eta + \eta' + \eta\eta')_1$, which is Morita equivalent to the trivial algebra $A = 1$ and does not lead to a new gauging. The subscripts "1, 2, 3" in the last line denote different discrete torsion / algebra structure choices, corresponding to the three fiber functors (or SPT phases) of Rep($D_4$).

Given these algebra objects, we can follow the universal steps in Section 2.2 to construct the

bipartite principal graphs of subfactors. This involves computing the reduced fusion matrix $F_A^r$, solving for $P_A^r$ satisfying $F^A = P_A^r (P_A^r)^{\mathrm{T}}$, and constructing the extended matrix $P_A$ by adding a column vector corresponding to the algebra object $A$. The principal graph is then the bipartite graph built from $P_A$ by labeling each row with a red vertex, each column with a black vertex, and constructing edges according to $(P_A)_{ij}$ between the $i$-th red vertex and the $j$-th black vertex. The red vertices correspond to simple objects of $\mathrm{Rep}(D_4)$ (even part of the subfactor), while the black vertices correspond to simple objects of the associated module category $\mathcal{M}(A)$ (odd part of the subfactor).

From the principal graphs, we can further construct quiver diagrams that encode representations of $\mathrm{Rep}(D_4)$ for a given algebra object $A$, following the method in Section 2.2. The quiver nodes correspond to the odd part of the principal graph, with each node representing a simple object $m_I$ in the module category. The links in the quiver diagram describe how these modules transform under a simple object $L_i$ in $\mathrm{Rep}(D_4)$, given by

$$L_i m_I = \sum_J N_{IJ}^i m_J, \tag{25}$$

which can be directly extracted from the principal graph.

Physically, these quiver diagrams have significant interpretations, particularly in the context of gapped phases described by 2D $\mathrm{Rep}(D_4)$-symmetric TQFTs. The classification of such gapped phases using the Symmetry TFT (SymTFT) was established in [BPSNW24], and the correspondence between these phases and gaugeable algebra objects in $\mathrm{Rep}(D_4)$ was identified in [PLRS$^+$24b]. This aligns with the general principle that every gaugeable algebra object $A$ in a fusion category $\mathcal{C}$ defines a $\mathcal{C}$-symmetric TQFT.

In this framework, the quiver diagram associated with a given $A$ describes how the topological boundary conditions $m_I$ of the $\mathrm{Rep}(D_4)$-symmetric TQFT transform under topological defect lines $L_i$, which form a fundamental part of the defining data for the TQFT associated with $A$ (see, e.g., [TW24a, HLS21]. Alternatively, the quiver diagram can be interpreted in terms of particle excitations in the vacuum labeled by $m_I$ and soliton configurations connecting different vacua $m_I$ and $m_J$. Particle-soliton degeneracy occurs when both types of excitations appear in a single quiver diagram. The results in this section thus provide a complete classification of particle-soliton degeneracies in $\mathrm{Rep}(D_4)$-symmetric gapped phases.

### 4.1    $A = 1$

The reduced fusion matrix $F_1^r$ and its associated $P_1^r$ matrix are given by

$$F_1^r = P_1^r = \begin{pmatrix} 1 & 0 & 0 & 0 \\ 0 & 1 & 0 & 0 \\ 0 & 0 & 1 & 0 \\ 0 & 0 & 0 & 1 \end{pmatrix}. \tag{26}$$

Enlarging $P_1^r$ by a column vector corresponding to the algebra object $(1, 0, 0, 0, 0)^{\mathrm{T}}$, we obtain

$$P_1 = \begin{pmatrix} 1 & 0 & 0 & 0 & 0 \\ 0 & 1 & 0 & 0 & 0 \\ 0 & 0 & 1 & 0 & 0 \\ 0 & 0 & 0 & 1 & 0 \\ 0 & 0 & 0 & 0 & 1 \end{pmatrix}. \tag{27}$$

The resulting bipartite principal graph is shown in Figure 3.



Figure 3: Principal graph for $\mathrm{Rep}(D_4)$ with $A = 1$.

From the principal graph, we see that for the trivial algebra object $A = 1$, the corresponding module category $\mathcal{M}_{(1)}$ is simply $\mathrm{Rep}(D_4)$ itself. Its simple objects can be labeled by the even parts they connect to:

$$\mathcal{M}_{(1)} : \{1, \eta, \eta', \eta\eta', \mathcal{N}\}. \tag{28}$$

The quiver diagrams for the representations of $\mathrm{Rep}(D_4)$ are determined by the fusion of simple objects in $\mathrm{Rep}(D_4)$ with the even parts to which each simple object in $\mathcal{M}_{(1)}$ is connected. The resulting quiver diagrams are shown in Figure 4. We omit the quiver diagrams for the $\eta'$ and $\eta\eta'$ representations, as they are identical to that of the $\eta$ representation, up to a relabeling of the quiver nodes associated with the invertible objects.

These quiver diagrams characterize the fully SSB phase of $\mathrm{Rep}(D_4)$ associated with $A = 1$ [BPSNW24].

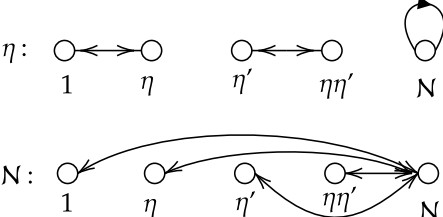

Figure 4: Quiver diagrams of representations of $\mathrm{Rep}(D_4)$ associated to $A = 1$.

There are five vacua, corresponding to the quiver nodes. In the $\eta$ representation, particle-soliton degeneracies appear, consisting of two soliton-anti-soliton pairs and a local excitation. In contrast, the $\mathcal{N}$ representation features only solitons.

## 4.2 $A = 1 + \eta$

The reduced fusion matrix $F^r_{1+\eta}$ and its associated $P^r_{1+\eta}$ matrix are given by

$$
F^r_{1+\eta} = \begin{pmatrix} 1 & 1 & 0 \\ 1 & 1 & 0 \\ 0 & 0 & 2 \end{pmatrix}, \quad P^r_{1+\eta} = \begin{pmatrix} 1 & 0 & 0 \\ 1 & 0 & 0 \\ 0 & 1 & 1 \end{pmatrix}. \tag{29}
$$

Enlarging $P^r_{1+\eta}$ by adding a column vector corresponding to the algebra object $(1, 1, 0, 0, 0)^{\mathrm{T}}$, we obtain

$$
P_{1+\eta} = \begin{pmatrix} 1 & 0 & 0 & 0 \\ 1 & 0 & 0 & 0 \\ 0 & 1 & 0 & 0 \\ 0 & 1 & 0 & 0 \\ 0 & 0 & 1 & 1 \end{pmatrix}. \tag{30}
$$

The resulting bipartite principal graph is shown in Figure 5.

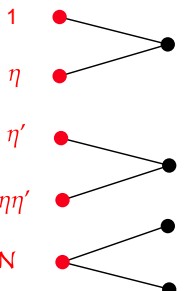

Figure 5: Principal graph of $A = 1 + \eta$.

From the principal graph, we identify four simple objects in the module category $\mathcal{M}_{(1+\eta)}$, labeled

by the even parts they connect to:

$$\mathcal{M}_{(1+\eta)} : \{1 + \eta, \eta' + \eta\eta', \mathcal{N}, \mathcal{N}\}. \tag{31}$$

Similarly, we can compute the principal graph for the algebra object $1 + \eta'$ (resp. $1 + \eta\eta'$). The resulting graph are identical to that in Figure 5, with the replacement $\eta \leftrightarrow \eta'$ (resp. $\eta \leftrightarrow \eta\eta'$). The corresponding module category objects are given by

$$\mathcal{M}_{(1+\eta')} : \{1 + \eta', \eta + \eta\eta', \mathcal{N}, \mathcal{N}\},$$
$$\mathcal{M}_{(1+\eta\eta')} : \{1 + \eta\eta', \eta + \eta', \mathcal{N}, \mathcal{N}\}. \tag{32}$$

The quiver diagrams for the representations of $\mathrm{Rep}(D_4)$ are determined by the fusion of simple objects in $\mathrm{Rep}(D_4)$ with the even parts to which each simple object in $\mathcal{M}_{(1+\eta)}$ is connected. The resulting quiver diagrams are shown in Figure 4. We omit the quiver diagram for the $\eta\eta'$ representation, as it is identical to that of the $\eta'$ representation, up to a relabeling of the quiver nodes associated with the invertible objects.

Figure 6: Quivers for representations of $\mathrm{Rep}(D_4)$ associated to the algebra object $1 + \eta$

These quiver diagrams characterize the $(\mathbb{Z}_2)^2$ SSB phase of $\mathrm{Rep}(D_4)$ associated with $A = 1+\eta$. The four vacua correspond to the quiver nodes, with no particle-soliton degeneracy. In the $\eta$ representation, only local particle excitations are present, while in the $\eta'$ and $\mathcal{N}$ representations, only solitons appear.

## 4.3 $A = 1 + \eta + \eta' + \eta\eta'$

This case is special in that it allows for two distinct gaugings. One gauging is equivalent to the trivial gauging $A = 1$, whose quantum symmetry is simply $\mathrm{Rep}(D_4)$ itself, while the other involves turning on a discrete torsion, resulting in a quantum symmetry $\mathrm{Vec}^\omega_{\mathbb{Z}_2^3}$ (see, e.g., [BT18]). Consequently, we expect to obtain two different subfactor principal graphs.

The reduced matrix in this case reads

$$F^r_{1+\eta+\eta'+\eta\eta'} = \begin{pmatrix} 0 & 0 & 0 & 0 \\ 0 & 0 & 0 & 0 \\ 0 & 0 & 0 & 0 \\ 0 & 0 & 0 & 4 \end{pmatrix}. \tag{33}$$

Compared to the previous two cases with $A = 1$ and $A = 1 + \eta$, this time there are two solutions for $P^r \cdot (P^r)^{\mathrm{T}} = F_r$, which we denote as $P^r_{(1+\eta+\eta'+\eta\eta')_1}$ and $P^r_{(1+\eta+\eta'+\eta\eta')_2}$, respectively:

$$P^r_{(1+\eta+\eta'+\eta\eta')_1} = \begin{pmatrix} 0 & 0 & 0 & 0 \\ 0 & 0 & 0 & 0 \\ 0 & 0 & 0 & 0 \\ 1 & 1 & 1 & 1 \end{pmatrix}, \quad P^r_{(1+\eta+\eta'+\eta\eta')_2} = \begin{pmatrix} 0 & 0 & 0 & 0 \\ 0 & 0 & 0 & 0 \\ 0 & 0 & 0 & 0 \\ 0 & 0 & 0 & 2 \end{pmatrix} \tag{34}$$

Enlarging these two solutions by adding a column vector corresponding to the algebra object $(1,1,1,1,0)^{\mathrm{T}}$, we have

$$P^r_{(1+\eta+\eta'+\eta\eta')_1} = \begin{pmatrix} 1 & 0 & 0 & 0 & 0 \\ 1 & 0 & 0 & 0 & 0 \\ 1 & 0 & 0 & 0 & 0 \\ 1 & 0 & 0 & 0 & 0 \\ 0 & 1 & 1 & 1 & 1 \end{pmatrix}, \quad P^r_{(1+\eta+\eta'+\eta\eta')_2} = \begin{pmatrix} 1 & 0 & 0 & 0 & 0 \\ 1 & 0 & 0 & 0 & 0 \\ 1 & 0 & 0 & 0 & 0 \\ 1 & 0 & 0 & 0 & 0 \\ 0 & 0 & 0 & 0 & 2 \end{pmatrix} \tag{35}$$

The resulting bipartite principal graphs are shown in Figure 7.

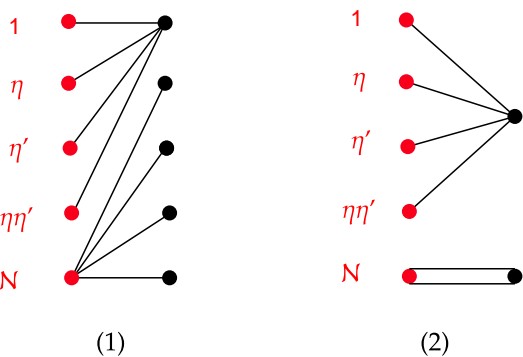

(1)                    (2)

Figure 7: Two principal graphs for the algebra object $1 + \eta + \eta' + \eta\eta'$.

From the principal graphs, we see that though the two gaugings are labeled by the same non-simple object $1 + \eta + \eta' + \eta\eta'$, they are indeed distinguished and lead to different module categories. The

simple objects of the two module categories are labeled by the even parts they connect to

$$\mathcal{M}_{(1+\eta+\eta'+\eta\eta')_1} : \{1+\eta+\eta'+\eta\eta', \mathcal{N}, \mathcal{N}, \mathcal{N}, \mathcal{N}\}$$

$$\mathcal{M}_{(1+\eta+\eta'+\eta\eta')_2} : \{1+\eta+\eta'+\eta\eta', 2\mathcal{N}\}.$$

(36)

At this stage, we can observe that the number of simple objects in $\mathcal{M}_{(1+\eta+\eta'+\eta\eta')_1}$ matches that in $\mathcal{M}_{(1)}$ as given in (28). We will establish that they are the same module category after building their quiver diagrams for representations.

The quiver diagrams for the representations of $\mathrm{Rep}(D_4)$ for these two principal graphs are determined by the fusion of simple objects of $\mathrm{Rep}(D_4)$ with the even parts to which each simple object in $\mathcal{M}_{(1+\eta+\eta'+\eta\eta')_{1,2}}$ is connected. The resulting quiver diagrams are shown in Figure 8. In both cases, we omit the quiver diagrams for the $\eta'$ and $\eta\eta'$ representations, as they are identical to that of the $\eta$ representation.

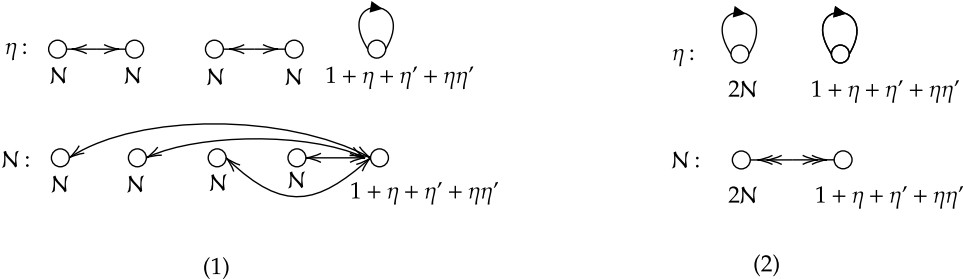

Figure 8: Quivers for representations of $\mathrm{Rep}(D_4)$ associated to $A = 1 + \eta$. The first one is equivalent to those of $A = 1$, implying the Morita-equivalence.

From quiver diagrams (1) in Figure 8, we observe that the quiver diagram for $\mathcal{M}_{(1+\eta+\eta'+\eta\eta')1}$ is identical to that of $\mathcal{M}_{(1)}$ in Figure 4, up to a relabeling of simple objects. We can see by definition that $A = (1 + \eta + \eta' + \eta\eta')_1$ and $A = 1$ are Morita equivalent, corresponding to the same gauging. Consequently, the associated 2D TQFT again describes the fully SSB phase of $\mathrm{Rep}(D_4)$, exhibiting particle-soliton degeneracy in the $\eta$ representation.

Quiver diagrams (2) in Figure 8 characterize the $\mathrm{Rep}(D_4)/(\mathbb{Z}_2 \times \mathbb{Z}_2)$ SSB phase associated with $A = (1 + \eta + \eta' + \eta\eta')_2$ [BPSNW24]. The two vacua correspond to the quiver nodes, with no particle-soliton degeneracy. In the $\eta$ representation, only local particle excitations are present, whereas in the $\mathcal{N}$ representation, only solitons appear.

## 4.4 $A = 1 + \eta + \mathcal{N}$

The reduced fusion matrix $F^r_{1+\eta+\mathcal{N}}$ and its associated $P^r_{1+\eta+\mathcal{N}}$ matrix are given by

$$F^r_{1+\eta+\mathcal{N}} = \begin{pmatrix} 0 & 0 & 0 & 0 \\ 0 & 1 & 1 & 1 \\ 0 & 1 & 1 & 1 \\ 0 & 1 & 1 & 1 \end{pmatrix}, \quad P^r_{1+\eta+\mathcal{N}} = \begin{pmatrix} 0 & 0 & 0 & 0 \\ 0 & 0 & 0 & 1 \\ 0 & 0 & 0 & 1 \\ 0 & 0 & 0 & 1 \end{pmatrix}. \tag{37}$$

Enpanding the $P^r_{1+\eta+\mathcal{N}}$ matrix by appending the column vector given by the algebra object $(1, 1, 0, 0, 1)^{\mathrm{T}}$, we have

$$P_{1+\eta+\mathcal{N}} = \begin{pmatrix} 1 & 0 & 0 & 0 & 0 \\ 1 & 0 & 0 & 0 & 0 \\ 0 & 0 & 0 & 0 & 1 \\ 0 & 0 & 0 & 0 & 1 \\ 1 & 0 & 0 & 0 & 1 \end{pmatrix}. \tag{38}$$

The resulting bipartite principal graph is shown in Figure 9.

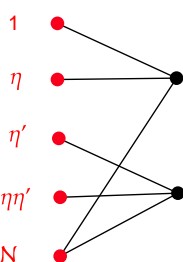

Figure 9: Principal graph for $A = 1 + \eta + \mathcal{N}$

From the graph, we identify two simple objects in the module category $\mathcal{M}_{(1+\eta+\mathcal{N})}$, labeled by the even parts they connect to:

$$\mathcal{M}_{(1+\eta+\mathcal{N})} : \{1 + \eta + \mathcal{N}, \eta' + \eta\eta' + \mathcal{N}\}. \tag{39}$$

Similarly, computing the principal graphs for the algebra objects $1 + \eta' + \mathcal{N}$ and $1 + \eta\eta' + \mathcal{N}$ yields graphs identical to Figure 9, with the replacement $\eta \leftrightarrow \eta'$ and $\eta \leftrightarrow \eta\eta'$, respectively. The corresponding module categories are

$$\mathcal{M}_{(1+\eta'+\mathcal{N})} : \{1 + \eta' + \mathcal{N}, \eta + \eta\eta' + \mathcal{N}\},$$
$$\mathcal{M}_{(1+\eta\eta'+\mathcal{N})} : \{1 + \eta\eta' + \mathcal{N}, \eta + \eta' + \mathcal{N}\}. \tag{40}$$

Without loss of generality, we will focus on the quivers on $\mathcal{M}_{(1+\eta+\mathcal{N})}$.

The quiver diagrams for representations are determined by the fusion of simple objects in $\mathrm{Rep}(D_4)$ with the even parts to which each simple object in $\mathcal{M}_{(1+\eta+\mathcal{N})}$ is connected. The resulting quiver diagrams are shown in Figure 10. We omit the quiver for the $\eta\eta'$ representation, as it is identical to that of the $\eta'$ representation.

Figure 10: Quiver representations for $A = 1 + \eta + \mathcal{N}$. Only $\mathcal{N}$ representation has particle-soliton degeneracy.

These quiver diagrams characterize the $\mathbb{Z}_2$ SSB phase of $\mathrm{Rep}(D_4)$ associated with $A = 1 + \eta + \mathcal{N}$ [BPSNW24]. The two vacua correspond to the quiver nodes, with particle-soliton degeneracy appearing in the $\mathcal{N}$ representation, which contains two particle states and a soliton-anti-soliton pair. In the $\eta$ representation, only local particle excitations are present, while in the $\eta'$ representation, only solitons appear.

## 4.5  $A = 1 + \eta + \eta' + \eta\eta' + 2\mathcal{N}$

These correspond to the maximal gaugings of $\mathrm{Rep}(D_4)$. In this case, the reduced fusion matrix and the associated $P^r_{1+\eta+\eta'+\eta\eta'+2\mathcal{N}}$ matrix are trivial:

$$F^r_{1+\eta+\eta'+\eta\eta'+2\mathcal{N}} = 0, \ P^r_{1+\eta+\eta'+\eta\eta'+2\mathcal{N}} = 0. \tag{41}$$

As a result, the principal graph consists of a single black vertex corresponding to the algebra object, as shown in Figure 11.

The corresponding module category $\mathcal{M}(1 + \eta + \eta' + \eta\eta' + 2\mathcal{N})$ contains only one simple object, labeled by the algebra object itself:

$$\mathcal{M}_{(1+\eta+\eta'+\eta\eta'+2\mathcal{N})} : \{1 + \eta + \eta' + \eta\eta' + 2\mathcal{N}\}. \tag{42}$$

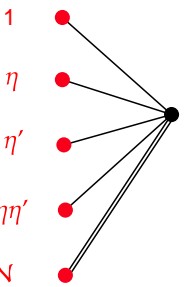

Figure 11: Principal graph of $A = 1 + \eta + \eta' + \eta\eta' + 2\mathcal{N}$.

The resulting quiver diagrams for the representations are shown in Figure 12. They characterize the SPT phases of $\mathrm{Rep}(D_4)$ associated with the maximal algebra object $1 + \eta + \eta' + \eta\eta' + 2\mathcal{N}$. The single quiver node corresponds to the unique vacuum for the SPT, with only particle excitations.

$$1,\ \eta,\ \eta',\eta\eta':$$

$$1 + \eta + \eta' + \eta\eta' + \mathsf{N}$$

$$\mathsf{N}:$$

$$1 + \eta + \eta' + \eta\eta' + \mathsf{N}$$

Figure 12: Quiver representations for $\mathrm{Rep}(D_4)$ associated to $A = 1 + \eta + \eta' + \eta\eta' + 2\mathcal{N}$. This describes a SPT phase described by $\mathrm{Rep}(D_4)$-symmetric TQFT.

To conclude this section, we summarize key aspects of gauging $\mathrm{Rep}(D_4)$ in Table 2, including the algebra objects, module categories, quantum symmetries as dual fusion categories ${}_{\mathcal{A}}\mathcal{C}_{\mathcal{A}}$, and particle-soliton degeneracies in the associated topological phases. The results in the first three columns were previously obtained in [DLWW24] using an alternative approach via NIM-reps.

| $A$ | $\mathcal{M}_{(A)}$ | ${}_{\mathcal{A}}\mathcal{C}_{\mathcal{A}}$ | Particle-soliton degeneracy? |
|---|---|---|---|
| $1$ | $\{1, \eta, \eta', \eta\eta', \mathcal{N}\}$ | $\mathrm{Rep}(D_4)$ | yes |
| $1 + \eta$ | $\{1 + \eta, \eta' + \eta\eta', \mathcal{N}, \mathcal{N}\}$ | $\mathrm{Vec}_{D_4}$ | no |
| $1 + \eta'$ | $\{1 + \eta', \eta + \eta\eta', \mathcal{N}, \mathcal{N}\}$ | $\mathrm{Vec}_{D_4}$ | no |
| $1 + \eta\eta'$ | $\{1 + \eta\eta', \eta + \eta', \mathcal{N}, \mathcal{N}\}$ | $\mathrm{Vec}_{D_4}$ | no |
| $(1 + \eta + \eta' + \eta\eta')_2$ | $\{1 + \eta + \eta' + \eta\eta', 2\mathcal{N}\}$ | $\mathrm{Vec}^{\omega}_{(\mathbb{Z}_2)^3}$ | no |
| $1 + \eta + \mathcal{N}$ | $\{1 + \eta + \mathcal{N}, \eta' + \eta\eta' + \mathcal{N}\}$ | $\mathrm{Rep}(D_4)$ | yes |
| $1 + \eta' + \mathcal{N}$ | $\{1 + \eta' + \mathcal{N}, \eta + \eta\eta' + \mathcal{N}\}$ | $\mathrm{Rep}(D_4)$ | yes |
| $1 + \eta\eta' + \mathcal{N}$ | $\{1 + \eta\eta' + \mathcal{N}, \eta + \eta' + \mathcal{N}\}$ | $\mathrm{Rep}(D_4)$ | yes |
| $(1 + \eta + \eta' + \eta\eta' + 2\mathcal{N})_1$ | $\{1 + \eta + \eta' + \eta\eta' + 2\mathcal{N}\}$ | $\mathrm{Vec}_{D_4}$ | no |
| $(1 + \eta + \eta' + \eta\eta' + 2\mathcal{N})_2$ | $\{1 + \eta + \eta' + \eta\eta' + 2\mathcal{N}\}$ | $\mathrm{Vec}_{D_4}$ | no |
| $(1 + \eta + \eta' + \eta\eta' + 2\mathcal{N})_3$ | $\{1 + \eta + \eta' + \eta\eta' + 2\mathcal{N}\}$ | $\mathrm{Vec}_{D_4}$ | no |

Table 2: Correspondence between gaugable algebra objects, module categories, dual categories in $Rep(D_4)$, and the presence / absence of particle-soliton degeneracy in the associated topological phases.

# 5 Rep($A_4$) and Non-invertible Triality

In this section, we examine the structure of Rep($A_4$) and its Morita-equivalent non-invertible triality symmetry $\mathcal{C}^{\mathcal{T}}$. The triality symmetry $\mathcal{C}^{\mathcal{T}}$ arises as the dual quantum symmetry when gauging a non-normal subgroup $\mathbb{Z}_2 \subset A_4$ [TW24b]. Equivalently, it can be obtained by gauging a specific algebra object (specified below) in Rep($A_4$) [PLRS$^+$24b]. These three symmetries form a Brauer-Picard groupoid, a generalized orbifold groupoid, indicating their interconnection via gauging.

While the group-theoretical data in [PLRS$^+$24b] classify all possible gaugings of $A_4$ and Rep($A_4$), the gaugings of the triality symmetry $\mathcal{C}^{\mathcal{T}}$ remain unexplored in the literature. Using the subfactor approach, we determine all possible gaugings of $\mathcal{C}^{\mathcal{T}}$ by examining potential dual principal graphs of subfactors corresponding to gaugings of $A_4$ and Rep($A_4$). This completes the Brauer-Picard groupoid for these three symmetries. We apply these results to construct global manipulations on the Kosterlitz-Thouless (KT) point of the $c = 1$ conformal manifold. Furthermore, for non-invertible Rep($A_4$) and $\mathcal{C}^{\mathcal{T}}$, we classify representations associated with all gaugeable algebra objects, providing a complete characterization of the gapped phases of these symmetries, as well as their particle-soliton degeneracies.

## 5.1 Rep($A_4$)

The alternating group $A_4$ of order 12 is defined by

$$\left\langle a, b \mid a^3 = b^2 = (ab)^3 = 1 \right\rangle, \tag{43}$$

where $a$ and $b$ are expressed in cycle notation as $a = (1, 2, 3)$ and $b = (1, 2)(3, 4)$.

Gauging the full invertible $A_4$ symmetry yields Rep($A_4$) as the dual quantum symmetry, comprising four simple objects: $1, \omega, \omega^2$, and $Z$, with the abelian fusion rules

$$\omega \times \omega^2 = 1, \ \omega \times Z = \omega^2 \times Z = Z, \ Z \times Z = 1 + \omega + \omega^2 + 2Z. \tag{44}$$

As a group-theoretical fusion category, the gaugings of Rep($A_4$) correspond one-to-one with those of $A_4$. There are seven distinct gaugings of Rep($A_4$), associated with the following algebra objects $A$

[PLRS+24b]:

$$1, \tag{45}$$

$$1 + \omega + \omega^2, \tag{46}$$

$$(1 + Z)_1, \ (1 + Z)_2, \tag{47}$$

$$1 + \omega + \omega^2 + Z, \tag{48}$$

$$(1 + \omega + \omega^2 + 3Z)_1, (1 + \omega + \omega^2 + 3Z)_2, (1 + \omega + \omega^2 + 3Z)_3. \tag{49}$$

The subscripts denote discrete torsion choices in gauging $(1 + Z)$ and $1 + \omega + \omega^2 + 3Z$, where the latter corresponds to two fiber functors (or SPT phases) of $\mathrm{Rep}(A_4)$.

### 5.1.1 Principal Graphs

Since we provided step-by-step derivations for subfactor principal graphs in the $\mathrm{Rep}(D_4)$ case, we directly list results for all algebra objects of $\mathrm{Rep}(A_4)$ here.

The reduced fusion matrices $F_A^r$ are given by

$$F_1^r = \begin{pmatrix} 1 & 0 & 0 \\ 0 & 1 & 0 \\ 0 & 0 & 1 \end{pmatrix}, F_{1+\omega+\omega^2}^r = \begin{pmatrix} 0 & 0 & 0 \\ 0 & 0 & 0 \\ 0 & 0 & 3 \end{pmatrix}, F_{1+Z}^r = \begin{pmatrix} 1 & 0 & 1 \\ 0 & 1 & 1 \\ 1 & 1 & 2 \end{pmatrix}, F_{1+\omega+\omega^2+Z}^r = \begin{pmatrix} 0 & 0 & 0 \\ 0 & 0 & 0 \\ 0 & 0 & 4 \end{pmatrix}, F_{1+\omega+\omega^2+3Z}^r = 0. \tag{50}$$

Their associated $P_A^r$ matrices, satisfying $P_A^r \cdot (P_A^r)^{\mathrm{T}} = F_A^r$, are:

$$P_1^r = \begin{pmatrix} 1 & 0 & 0 \\ 0 & 1 & 0 \\ 0 & 0 & 1 \end{pmatrix}, P_{1+\omega+\omega^2}^r = \begin{pmatrix} 0 & 0 & 0 \\ 0 & 0 & 0 \\ 1 & 1 & 1 \end{pmatrix}, P_{1+Z}^r = \begin{pmatrix} 1 & 0 \\ 0 & 1 \\ 1 & 1 \end{pmatrix}, P_{1+\omega+\omega^2+Z}^r = \begin{pmatrix} 0 \\ 0 \\ 2 \end{pmatrix}, P_{1+\omega+\omega^2+3Z}^r = 0. \tag{51}$$

Enlarging these $P_A^r$ matrices by appending the column vector given by the algebra object, e.g. $(1, 1, 1, 3)^{\mathrm{T}}$ for $A = 1 + \omega + \omega^2 + 3Z$, we have the resulting bipartite graphs shown in Figure 13. Although $(1 + Z)$ and $(1 + \omega + \omega^2 + 3Z)$ admit multiple gaugings with discrete torsion choices, their corresponding principal graphs remain identical.

From the principal graphs, we identify simple objects in the module categories $\mathcal{M}_{(A)}$ based on their

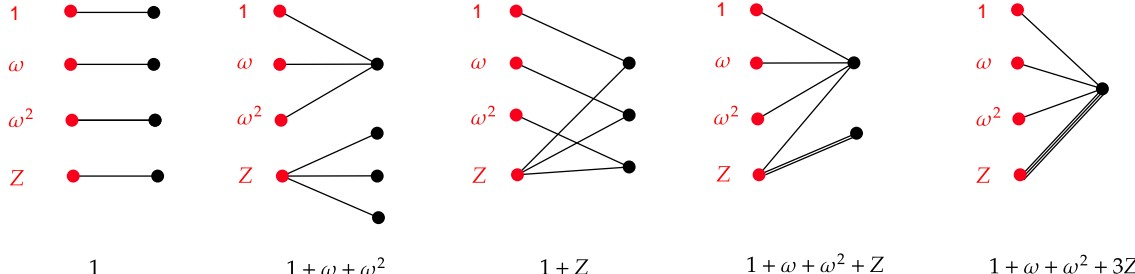

Figure 13: Principal graphs for $\text{Rep}(A_4)$.

connections to even parts:

$$\mathcal{M}_{(1)} : \{1, \omega, \omega^2, Z\},$$

$$\mathcal{M}_{(1+\omega+\omega^2)} : \{1 + \omega + \omega^2, Z, Z, Z\},$$

$$\mathcal{M}_{(1+Z)} : \{1 + Z, \omega + Z, \omega^2 + Z\}, \tag{52}$$

$$\mathcal{M}_{(1+\omega+\omega^2+Z)} : \{1 + \omega + \omega^2 + Z, 2Z\},$$

$$\mathcal{M}_{(1+\omega+\omega^2+3Z)} : \{1 + \omega + \omega^2 + 3Z\}.$$

The quiver diagrams for representations are determined by the fusion of simple objects in $\text{Rep}(A_4)$ with the even parts to which each simple objects in $\mathcal{M}_{(A)}$ is connected. The resulting quiver diagrams are summarized in Figure 14. We omit the quivers for the $\omega'$ representation, as it is identical to that of $\omega$ in all cases.

### 5.1.2 Classifying gapped phases and particle-soliton degeneracies

The gapped phases associated with $\text{Rep}(A_4)$ symmetry correspond one-to-one with algebra objects $A$. The quiver diagrams we obtain in Figure 14 characterize the vacua structure, where each quiver node gives rise to a vacuum, and links label how vacua transform under the $\text{Rep}(A_4)$ symmetry. Furthermore, the quiver links also classify the particle excitations and solitons upon these vacua, where the link connecting a single node gives a particle state, while a link connecting two quiver nodes labels a soliton. The results are summarized below:

- $A = 1$: $\text{Rep}(A_4)$ SSB phase. The quiver nodes represent the four vacua, with particle-soliton degeneracy occurring in both $\omega$ and $Z$ representations.

- $A = 1 + \omega + \omega^2$: 4-vacua SSB phase. The quiver nodes correspond to the four vacua, with no particle-soliton degeneracy.

- $A = 1 + Z$: Two 3-vacua SSB phases (associated with discrete torsion choices $(1 + Z)_1$ and $(1 + Z)_2$). In each SSB phase, the quiver nodes correspond to the three vacua, with particle-

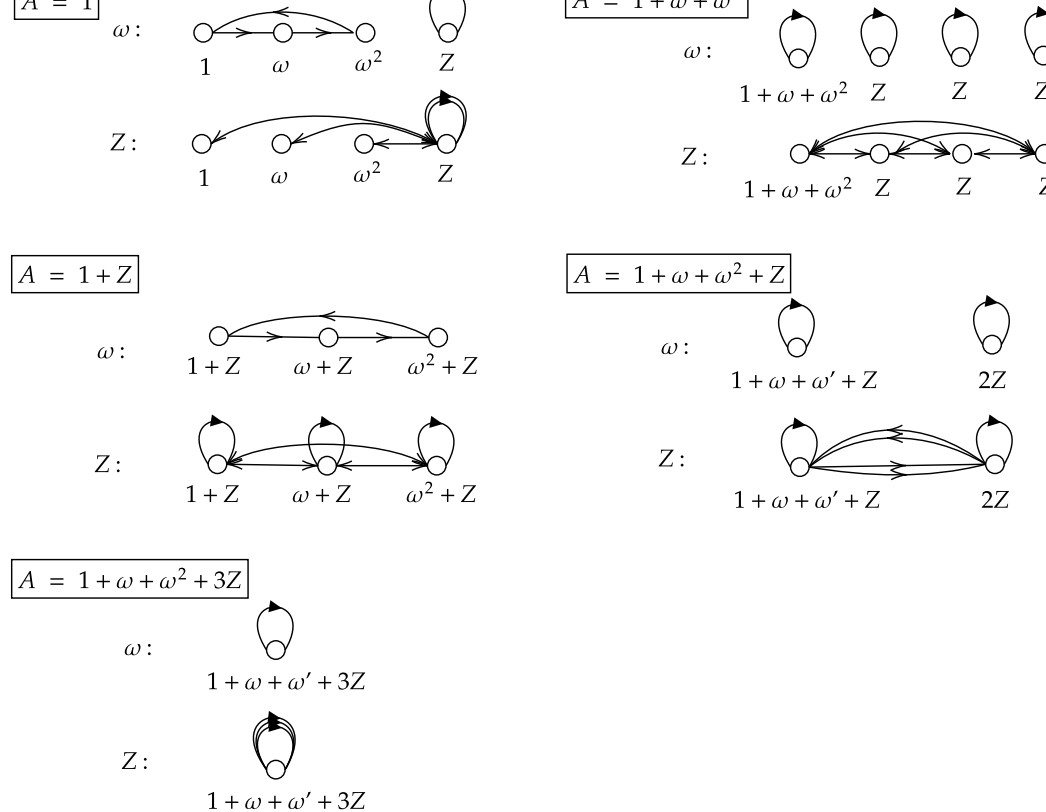

Figure 14: Quiver diagrams for representations of $\mathrm{Rep}(A_4)$. These diagrams also describe the gapped phases with $\mathrm{Rep}(A_4)$ symmetry.

soliton degeneracy in the $Z$ representation.

- $A = 1 + \omega + \omega^2 + Z$: 2-vacua SSB phase. The quiver nodes correspond to two vacua, with particle-soliton degeneracy in the $Z$ representation.

- $A = 1 + \omega + \omega^2 + 3Z$: Two $\mathrm{Rep}(A_4)$ SPT phases (associated with the two fiber functors of $\mathrm{Rep}(A_4)$). The single quiver node represents the unique vacuum for each SPT phase, supporting only particle excitations.

## 5.2  Non-invertible triality $\mathcal{C}^{\mathcal{T}}$

The non-invertible triality $\mathcal{C}^{\mathcal{T}}$ is realized in [TW24b] within the context of $c = 1$ CFTs by gauging a $\mathbb{Z}_2 \subset A_4$ symmetry of the $SU(2)_1$ theory. This category consists of six simple objects: $1, \eta, \eta', \eta\eta', \mathcal{T}$, and its orientation-reversed counterpart $\bar{\mathcal{T}}$. Their fusion rules are given by

$$\eta \times \eta = \eta' \times \eta' = 1, \quad \mathcal{T} \times \bar{\mathcal{T}} = 1 + \eta + \eta' + \eta\eta',$$

$$\mathcal{T} \times \eta = \mathcal{T} \times \eta' = \mathcal{T}, \quad \bar{\mathcal{T}} \times \eta = \bar{\mathcal{T}} \times \eta' = \bar{\mathcal{T}}, \tag{53}$$

$$\mathcal{T} \times \mathcal{T} = 2\bar{\mathcal{T}}, \quad \bar{\mathcal{T}} \times \bar{\mathcal{T}} = 2\mathcal{T}.$$

Alternatively, in [PLRS$^+$24b], this symmetry is also realized as the dual quantum symmetry obtained by gauging the algebra object $A = 1 + \omega + \omega^2 + Z$ in $\text{Rep}(A_4)$.

### 5.2.1  Classifying gaugings of $\mathcal{C}^{\mathcal{T}}$ via subfactors

We now explore how to gauge the $\mathcal{C}^{\mathcal{T}}$ symmetry using subfactors. Recall that for a fusion category $\mathcal{C}$ with an algebra object $A$ described by a principal graph, the dual principal graph in the subfactor corresponds to the dual fusion category $\mathcal{C}'$, consisting of $A$-$A$ bimodules over $\mathcal{C}$ and equipped with an algebra object $A'$. From a symmetry perspective, $\mathcal{C}'$ arises as the quantum symmetry obtained by gauging $A$ in $\mathcal{C}$, while $\mathcal{C}$ is the quantum symmetry from gauging $A'$ in $\mathcal{C}'$. The dimensions of the algebra objects $A$ and $A'$ coincide with the index of the subfactors and the module categories.

Furthermore, the odd part of the principal graph, which labels modules over $\mathcal{C}$, should have a one-to-one correspondence with the odd part of the dual principal graph, which labels modules over $\mathcal{C}'$. Physically, this correspondence of odd parts is interpreted as the topological interface arising from half-space gauging of $A$ in $\mathcal{C}$ or, equivalently, half-space gauging of $A'$ in $\mathcal{C}'$.

Building on this principle, classifying nontrivial gaugings of the triality symmetry $\mathcal{C}^{\mathcal{T}}$ involves identifying subfactors whose principal graphs correspond to the dual principal graphs for gauging $\mathbb{Z}_2 \subset A_4$, as well as for gauging $1 + \omega + \omega^2 + Z$ in $\text{Rep}(A_4)$. The former has a subfactor index of 2, corresponding to the fact that the algebra object for gauging $\mathbb{Z}_2$ has dimension 2. Given the dimensions of the simple objects in $\mathcal{C}^{\mathcal{T}}$:

$$\langle 1 \rangle = \langle \eta \rangle = \langle \eta' \rangle = \langle \eta\eta' \rangle = 1, \ \langle \mathcal{T} \rangle = \langle \bar{\mathcal{T}} \rangle = 2, \tag{54}$$

we find three possible principal graphs with index 2, corresponding to the algebra objects $1 + \eta$, $1 + \eta'$, and $1 + \eta\eta'$ in $\mathcal{C}^{\mathcal{T}}$. The principal graph for $1 + \eta$ is illustrated in Figure 15, while those for $1 + \eta'$ and $1 + \eta\eta'$ are omitted, as they are identical to that of $1 + \eta$ with $\eta \leftrightarrow \eta'$ and $\eta \leftrightarrow \eta\eta'$, respectively.

Similarly, we can explore the principal graphs whose dual principal graph corresponds to the gauging of $1 + \omega + \omega^2 + Z$ in $\text{Rep}(A_4)$, as shown in the fourth image of Figure 13. The index of the corresponding subfactor is $\langle 1 + \omega + \omega^2 + Z \rangle = 6$. The potential algebra objects $A$ of $\mathcal{C}^{\mathcal{T}}$ with the correct dimension,

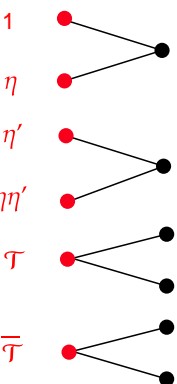

Figure 15: Principal graph of $1 + \eta$ for triality.

up to exchanging the three nontrivial invertible objects $\eta$, $\eta'$, and $\eta\eta'$, are:

$$1 + \eta + 2\mathcal{T}, \ 1 + \eta + 2\bar{\mathcal{T}}, 1 + \eta + \eta' + \eta\eta' + \mathcal{T}, \ 1 + \eta + \eta' + \eta\eta' + \bar{\mathcal{T}}, \ 1 + \eta + \mathcal{T} + \bar{\mathcal{T}}. \tag{55}$$

Next, we compute the reduced fusion matrices $F_A^r$ for these potential algebra objects $A$, and solve for the $P_A^r$ matrix satisfying $F_A^r = P_A^r \cdot (P_A^r)^{\mathrm{T}}$. It turns out that the only solution from (55) that yields non-negative integer elements for $P_A^r$ is $A = 1 + \eta + \mathcal{T} + \bar{\mathcal{T}}$, with the reduced fusion matrix and the corresponding $P_A^r$-matrix given by:

$$F_{1+\eta+\mathcal{T}+\bar{\mathcal{T}}}^r = \begin{pmatrix} 0 & 0 & 0 & 0 & 0 \\ 0 & 1 & 1 & 1 & 1 \\ 0 & 1 & 1 & 1 & 1 \\ 0 & 1 & 1 & 1 & 1 \\ 0 & 1 & 1 & 1 & 1 \end{pmatrix}, \ P_{1+\eta+\mathcal{T}+\bar{\mathcal{T}}}^r = \begin{pmatrix} 0 \\ 1 \\ 1 \\ 1 \\ 1 \end{pmatrix}. \tag{56}$$

The resulting principal graph is shown in Figure 16. By exchanging $\eta \leftrightarrow \eta'$ and $\eta \leftrightarrow \eta\eta'$, we can construct principal graphs for the other two gaugings dual to $\mathrm{Rep}(A_4)$.

Together with the trivial gauging $A = 1$, whose dual quantum symmetry is simply $\mathcal{C}^{\mathcal{T}}$ itself, we now have seven distinct gaugings for $\mathcal{C}^{\mathcal{T}}$, which matches the number of gaugings for $A_4$ and $\mathrm{Rep}(A_4)$ as expected. This is because they are Morita-equivalent within the same Brauer-Picard groupoid (see [EGNO15] for more details). We summarize this information about the generalized orbifold groupoid by illustrating the correspondence between the gaugings of $A_4$, $\mathrm{Rep}(A_4)$, and $\mathcal{C}^{\mathcal{T}}$ in Table 3.

### 5.2.2 Global manipulations on $c = 1$ Kosterlitz-Thouless CFT

Having identified all gaugings of the triality symmetry $\mathcal{C}^{\mathcal{T}}$, we now apply these results to an explicit 2D theory: the Kosterlitz-Thouless point of $c = 1$ CFTs, where this triality symmetry was first derived

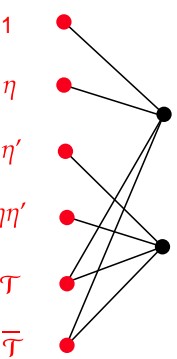

Figure 16: Principal graph of $A = 1 + \eta + \mathcal{T} + \bar{\mathcal{T}}$, whose dual principal graph is that of $\mathrm{Rep}(A_4)$ with $A = 1 + \omega + \omega' + Z$.

| $H \subset A_4$ | $A$ of $\mathrm{Rep}(A_4)$ | $A$ of $\mathcal{C}^T$ | $_A\mathcal{C}_A$ |
|---|---|---|---|
| $1$ | $(1 + \omega + \omega^2 + 3Z)_1$ | $1 + \eta$ | $\mathrm{Vec}_{A_4}$ |
| $\mathbb{Z}_2$ | $1 + \omega + \omega^2 + Z$ | $1$ | $\mathcal{C}^T$ |
| $\mathbb{Z}_3$ | $(1 + Z)_1$ | $1 + \eta + \mathcal{T} + \bar{\mathcal{T}}$ | $\mathrm{Rep}(A_4)$ |
| $(\mathbb{Z}_2)^2$ | $1 + \omega + \omega^2$ | $1 + \eta'$ | $\mathrm{Vec}_{A_4}$ |
| $(\mathbb{Z}_2)^2_{\mathrm{d.t.}}$ | $(1 + \omega + \omega^2 + 3Z)_2$ | $1 + \eta\eta'$ | $\mathrm{Vec}_{A_4}$ |
| $A_4$ | $1$ | $1 + \eta' + \mathcal{T} + \bar{\mathcal{T}}$ | $\mathrm{Rep}(A_4)$ |
| $(A_4)_{\mathrm{d.t.}}$ | $(1 + Z)_2$ | $1 + \eta\eta' + \mathcal{T} + \bar{\mathcal{T}}$ | $\mathrm{Rep}(A_4)$ |

Table 3: Correspondence between algebra objects of gauging non-invertible triality $\mathcal{C}^T$ and gauging $\mathrm{Rep}(A_4)$, as well as the dual quantum symmetries.

in [TW24a]. On the conformal manifold of the $c = 1$ CFT, the Kosterlitz-Thouless point is where the circle branch and the orbifold branch meet. Gaugings of triality $\mathcal{C}^T$ correspond to global manipulations of the $c = 1$ conformal manifold, mapping the Kosterlitz-Thouless point to other points.

To determine the resulting theories under these global manipulations, recall that the Kosterlitz-Thouless CFT is obtained by gauging a $\mathbb{Z}_2$ symmetry of the $SU(2)_1$ CFT, which represents the self-dual point under T-duality on the circle branch. As an $A_4$-symmetric theory, all possible $H \subset A_4$ gaugings for the $SU(2)_1$ theory are known [TW24b]. The theories that arise under gauging $H \subset A_4$ are:

$$H = 1 : SU(2)_1,$$

$$H = \mathbb{Z}_2 : \text{Kosterlitz-Thouless CFT},$$

$$H = \mathbb{Z}_3 : SU(2)_1/\mathbb{Z}_3 \text{ at } R = 3 \text{ on the circle branch}, \tag{57}$$

$$H = (\mathbb{Z}_2)^2 \text{ and } (\mathbb{Z}_2)^2_{\mathrm{d.t.}} : SU(2)_1/(\mathbb{Z}_2)^2 \text{ at } R = 2 \text{ on the orbifold branch},$$

$$H = A_4 \text{ and } (A_4)_{\mathrm{d.t.}} : SU(2)_1/A_4 \text{ at an isolated point of the conformal manifold}.$$

While generally turning on discrete torsion leads to distinct theories, in this case, due to the 't Hooft anomaly for the $(SU(2) \times SU(2))/\mathbb{Z}_2 \cong SO(4)$ global symmetries, turning on discrete torsion

physically gives the same theory [TW24b].

We now utilize the correspondence between the gaugings of $\mathcal{C}^{\mathcal{T}}$ and $A_4$ gaugings, as summarized in Table 3, to track how gaugings of $\mathcal{C}^{\mathcal{T}}$ topologically manipulate the Kosterlitz-Thouless CFT into other theories on the $c = 1$ conformal manifold. The result is illustrated in Figure 17.

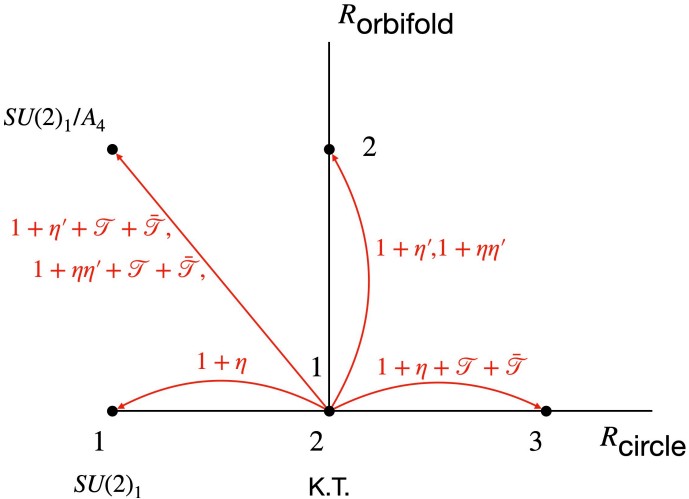

Figure 17: Global manipulations on K.T. CFT via gauging the triality.

### 5.2.3 Classfying gapped phases and particle-soliton degeneracies

After discussing the applications of gauging $\mathcal{C}^{\mathcal{T}}$ to the $c = 1$ conformal manifold for gapless phases, we now turn to its application in gapped phases described by $\mathcal{C}^{\mathcal{T}}$-symmetric TQFTs. The gapped phases correspond one-to-one to the algebra objects $A$ listed in Table 3.

As explored in the cases of $\mathrm{Rep}(D_4)$ and $\mathrm{Rep}(A_4)$, the vacuum structure of gapped phases is encoded in quiver diagrams, which describe representations of noninvertible symmetries. These diagrams are derived from the module category information in subfactor principal graphs. The principal graphs for $A = 1 + \eta$ and $A = 1 + \eta + \mathcal{T} + \bar{\mathcal{T}}$ are already provided in Figures 15 and 16. Additionally, there is a principal graph for the trivial algebra object $A = 1$, with the reduced fusion matrix $F_1^r$ and its associated $P_1^r$ matrix given by

$$F_1^r = P_1^r = \begin{pmatrix} 1 & 0 & 0 \\ 0 & 1 & 0 \\ 0 & 0 & 1 \end{pmatrix}. \tag{58}$$

The resulting principal graph is shown in Figure 18.

From the principal graphs in Figures 15, 16, and 18, we obtain the module categories with simple



Figure 18: Principal graph with trivial algebra object $A = 1$.

objects labeled by the even parts they connect to:

$$\mathcal{M}_{(1)} : \{1, \eta, \eta', \eta\eta', \mathcal{T}, \bar{\mathcal{T}}\},$$

$$\mathcal{M}_{(1+\eta)} : \{1 + \eta, \eta' + \eta\eta', \mathcal{T}, \mathcal{T}, \bar{\mathcal{T}}, \bar{\mathcal{T}}\}, \tag{59}$$

$$\mathcal{M}_{(1+\eta+\mathcal{T}+\bar{\mathcal{T}})} : \{1 + \eta + \mathcal{T} + \bar{\mathcal{T}}, \eta' + \eta\eta' + \mathcal{T} + \bar{\mathcal{T}}\}.$$

The quiver diagrams for representations are determined by the fusion of simple objects in $\mathcal{C}^{\mathcal{T}}$ with the even parts to which each simple object in $\mathcal{M}_A$ is connected. The resulting quiver diagrams are summarized in Figure 19. We omit the $\eta'$ representation in the $A = 1$ case, as it is identical to that of $\eta$ by relabeling the quiver nodes. The $\bar{\mathcal{T}}$ representations for all cases can be obtained from those of $\mathcal{T}$ by exchanging $\mathcal{T} \leftrightarrow \bar{\mathcal{T}}$ in the quiver diagrams.

From the quiver diagrams, we observe that no gapped phase has a unique ground state, leading to the conclusion that there is no SPT phase for $\mathcal{C}^{\mathcal{T}}$. This result is consistent with [TW24b] (see also [LS23]), where the authors proposed that the triality symmetry $\mathcal{C}^{\mathcal{T}}$ is anomalous, as the Kosterlitz-Thouless point in $c = 1$ CFT does not admit an SPT phase in the IR. All $\mathcal{C}^{\mathcal{T}}$-symmetric TQFTs, therefore, describe SSB phases, whose vacuum structure and particle-soliton degeneracies can be extracted from the quiver diagrams in Figure 19. We summarize the results in Table 4.

| $A$ of $\mathcal{C}^{\mathcal{T}}$ | $\mathcal{M}_{(A)}$ | Gapped phase | Particle-soliton degeneracy? |
|---|---|---|---|
| $1$ | $\{1, \eta, \eta', \eta\eta', \mathcal{T}, \bar{\mathcal{T}}\}$ | $\mathcal{C}^{\mathcal{T}}$ SSB | yes |
| $1 + \eta$ | $\{1 + \eta, \eta' + \eta\eta', \mathcal{T}, \mathcal{T}, \bar{\mathcal{T}}, \bar{\mathcal{T}}\}$ | 6-vacuum SSB$_1$ | no |
| $1 + \eta'$ | $\{1 + \eta', \eta + \eta\eta', \mathcal{T}, \mathcal{T}, \bar{\mathcal{T}}, \bar{\mathcal{T}}\}$ | 6-vacuum SSB$_2$ | no |
| $1 + \eta\eta'$ | $\{1 + \eta\eta', \eta + \eta\eta', \mathcal{T}, \mathcal{T}, \bar{\mathcal{T}}, \bar{\mathcal{T}}\}$ | 6-vacuum SSB$_3$ | no |
| $1 + \eta + \mathcal{T} + \bar{\mathcal{T}}$ | $\{1 + \eta + \mathcal{T} + \bar{\mathcal{T}}, \eta' + \eta\eta' + \mathcal{T} + \bar{\mathcal{T}}\}$ | 2-vacua SSB$_1$ | yes |
| $1 + \eta' + \mathcal{T} + \bar{\mathcal{T}}$ | $\{1 + \eta' + \mathcal{T} + \bar{\mathcal{T}}, \eta + \eta\eta' + \mathcal{T} + \bar{\mathcal{T}}\}$ | 2-vacua SSB$_2$ | yes |
| $1 + \eta\eta' + \mathcal{T} + \bar{\mathcal{T}}$ | $\{1 + \eta\eta' + \mathcal{T} + \bar{\mathcal{T}}, \eta + \eta' + \mathcal{T} + \bar{\mathcal{T}}\}$ | 2-vacua SSB$_3$ | yes |

Table 4: Classification of gapped phases described by $\mathcal{C}^{\mathcal{T}}$-symmetric TQFTs and particle-soliton degeneracies.

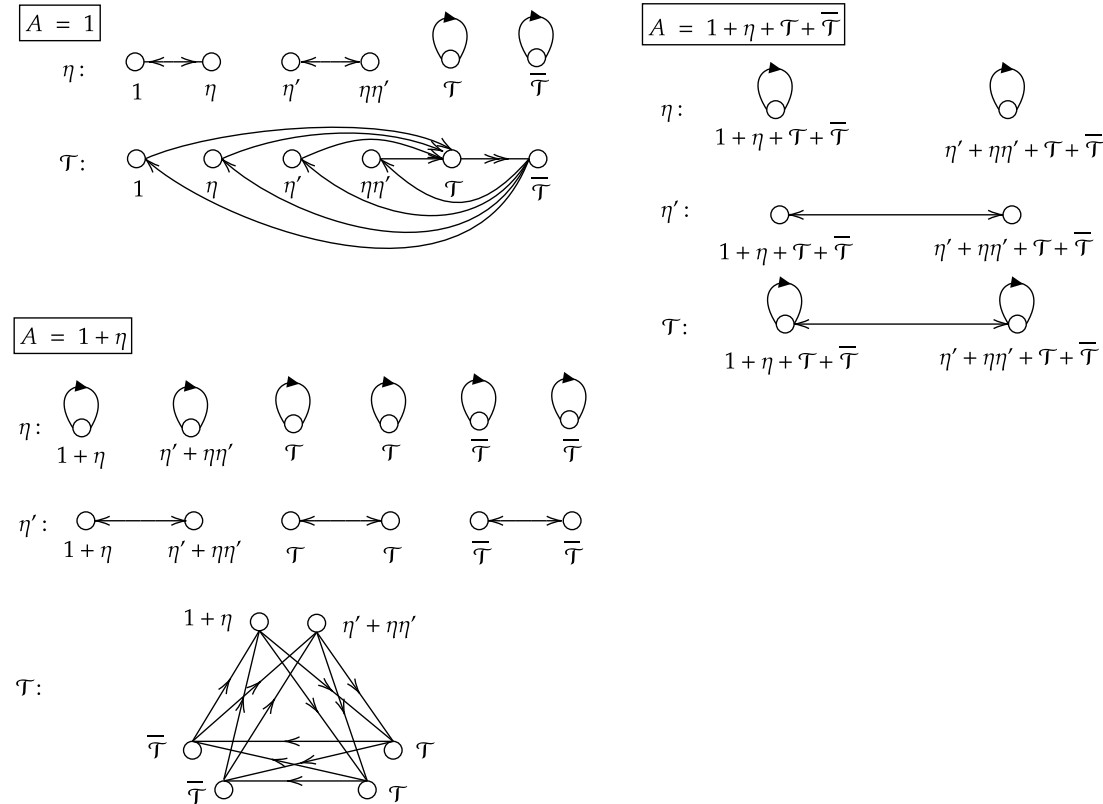

Figure 19: Quivers for representations of the non-invertible triality. There is no quiver with a single node, matching the fact that the symmetry is anomalous (no SPT phase).

# 6    $\mathbf{Rep}(A_5)$

In this section, we examine the non-invertible symmetry $\mathrm{Rep}(A_5)$, the representation category of the smallest non-solvable finite group $A_5$. We determine all gaugeable algebra objects in $\mathrm{Rep}(A_5)$ and construct the corresponding principal graphs and quiver diagrams for its modules and representations. We classify the dual quantum symmetries arising from all possible gaugings as fusion categories of $H$-$H$ bimodules over $A_5$. Since $\mathrm{Rep}(A_5)$ has no normal subgroups other than 1 and $A_5$ itself, all its Morita-equivalent categories—except for $\mathrm{Vec}_{A_5}$—are non-invertible.

We explore physical implications of these results for both CFTs and TQFTs. In the context of $c = 1$ CFTs, we analyze the exceptional $SU(2)_1/A_5$ theory, illustrating how gauging $\mathrm{Rep}(A_5)$ induces global manipulations on its conformal manifold. Notably, we identify a self-duality of $SU(2)_1/A_5$ under $\mathrm{Rep}(A_5)$ gauging and realize an extended non-invertible $\mathrm{Rep}(SL(2,5))$ symmetry via the self-duality. In the context of $\mathrm{Rep}(A_5)$-symmetric TQFTs, we use quiver diagrams to characterize the vacuum structure of the associated gapped phases and classify their particle-soliton degeneracies.

## 6.1 Subfactors and gaugeable algebra objects

The alternating $A_5$ of order 60 is defined by even permutations on five elements. It is the smallest non-solvable finite group, i.e. its only normal subgroup is the trivial group and itself. Its has five irreducible representations, which we denote as $1, X, X', U, V$ with dimensions $1, 3, 3, 4$ and $5$ respectively. These representations give rise to the five simple objects in the fusion category $\mathrm{Rep}(A_5)$, with the following abelian fusion rules (see also [TW24b]):

$$X \times X = 1 + X + V, \ X' \times X' = 1 + X' + V, \ X \times X' = U + V,$$

$$X \times U = X' + U + V, \ X' \times U = X + U + V, \ X \times V = X' \times V = X + X' + U + V, \tag{60}$$

$$U \times U = 1 + X + X' + U + V, \ U \times V = X + X' + U + 2V, \ V \times V = 1 + X + X' + 2U + 2V.$$

As discussed in, e.g., [BT18, PLRS$^+$24a, DLWW24], the algebra objects $A$ for a $\mathrm{Rep}(G)$ symmetry have a one-to-one correspondence to the gaugeable subgroup $H \subset G$ with discrete torsion choices. In particular, the modules associated to $A$ can be identified with those of $\mathrm{Vec}(A_4)$, labeled by $H$ and the discrete torsion. More explicitly, the simple objects in the $A$-module category over $\mathrm{Rep}(G)$ should associate with the (projective) representations of $H$. Despite its conceptual clarity, concrete computation for $A$ can be subtle.

Luckily, subfactors provide a universal method for the derivation of algebra objects. For a general candidate $A = 1 + a_1 X + a_2 X' + a_3 U + a_4 V$, we can derive its reduced fusion matrix $F_A^r$ and solving $P_A^r$ for $P_A^r \cdot (P_A^r)^{\mathrm{T}} = F_A^r$. If there is no non-negative integer solution for $P_A^r$, then $A$ is not an legal algebra object. If a non-negative integer solution exists, one can then draw the resulting principal graph and investigate its odd parts as the module category associated to $A$. Reading the number of odd parts and the dimensions of even parts they connect to, one can try to identify if there is a $H \subset G$, whose number and dimension ratio of (projective) representations match those of odd parts in the principal graph. If there is no such $H$ exist, $A$ can also be excluded. Searching all possible candidates $A$, with upper bound $a_i \leq d_i$ where $d_i$ is the dimension of the $i-th$ simple object in $\mathrm{Rep}(G)$, we can construct all gaugeable algebra objects.

Let us take the algebra object $1 + X$ of $\mathrm{Rep}(A_5)$ as an example to illustrate the above general strategy. The reduced fusion matrix $F_{1+X}^r$ and its associated $P_{1+X}^r$ are given by

$$F_{1+X}^r = \begin{pmatrix} 1 & 0 & 0 & 1 \\ 0 & 1 & 1 & 1 \\ 0 & 1 & 2 & 1 \\ 1 & 1 & 1 & 2 \end{pmatrix}, \ P_{1+X}^r = \begin{pmatrix} 1 & 0 & 0 \\ 0 & 1 & 0 \\ 0 & 1 & 1 \\ 1 & 1 & 0. \end{pmatrix} \tag{61}$$

The resulting principal graph is shown in Figure 20. There are four vertices in the odd part, corre-

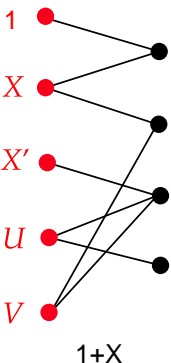

Figure 20: Principal graph of $1 + X$.

sponding to the four simple objects in the module category, which can be labeled by the even parts they connect to:

$$\mathcal{M}_{(1+X)} : \{1 + X, \ X + V, \ X' + U + V, \ U\} \tag{62}$$

The ratio of dimensions are given by

$$\langle 1 + X \rangle : \langle X + V \rangle : \langle X' + U + V \rangle : \langle U \rangle = 1 : 2 : 3 : 1. \tag{63}$$

This matches perfectly the four projective representations of $A_5$, whose dimensions are $2, 4, 6$ and $2$. Therefore, we have following correspondence between the $A_5$ gauging and the $\mathrm{Rep}(A_5)$ gaugeable algebra object:

$$(A_5)_{\mathrm{d.t.}} \longleftrightarrow 1 + X \text{ of } \mathrm{Rep}(A_5) \tag{64}$$

namely gauging $A_5$ with discrete torsion turned on.

Using this method, we obtain all gaugeable algebra objects, up to Morita-equivalence, for $\mathrm{Rep}(A_5)$, whose principal graphs are shown in Figure 21.

There are three other algebra objects labeling well-defined gauging for $\mathrm{Rep}(A_5)$

$$1 + X', 1 + X + X' + U + V, \ 1 + 2X + 2X' + 2U + 3V, \tag{65}$$

but they are Morita-equivalent to $A = 1 + X$ gauging, so we omit their principal graphs and leave them as an exercise for the interested reader. We summarize all possible gaugings of $\mathrm{Rep}(A_5)$, their associated $H \subset G$ gaugings, and dual quantum symmetries in Table 5. From the table, we find $A_5$, $\mathrm{Rep}(A_5)$, and other nine non-invertible symmetries connected via gauging, thus eleven fusion categories as objects in the Brauer-Picard groupoid of $\mathrm{Rep}(A_5)$. The dual fusion categories are labeled with the group-theoretical notation $\mathcal{C}(A_5, 1; H, \sigma)$ where $H$ is a subgroup of $G$ and $\sigma$ labels possible discrete

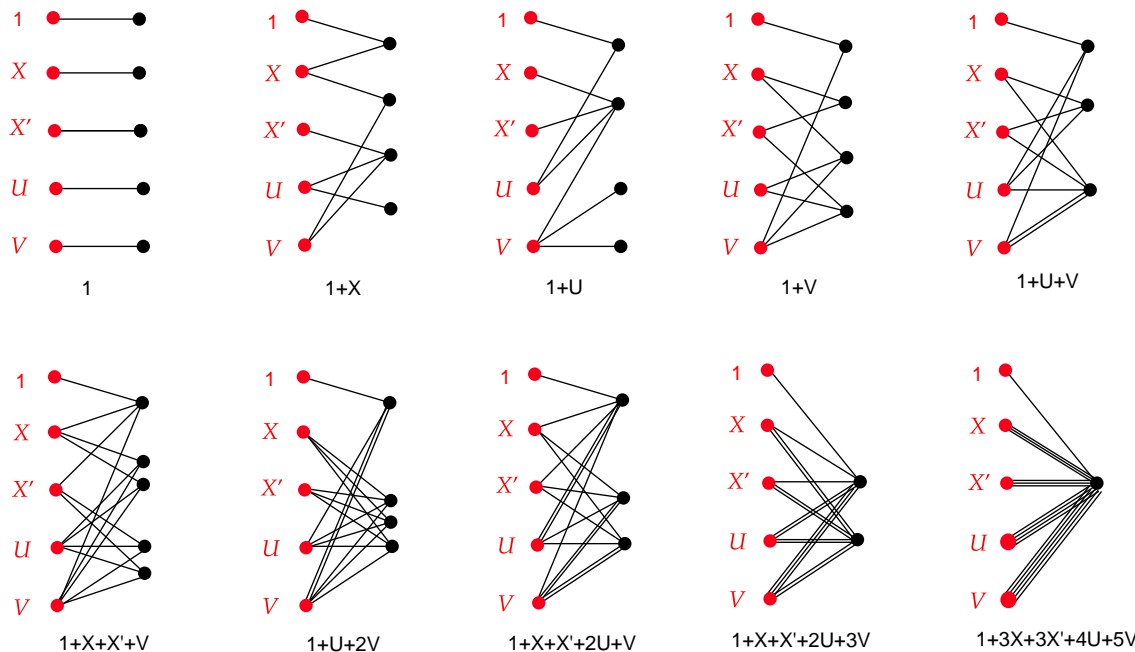

Figure 21: Principal graphs for all gaugeable algebra objects, up to Morita-equivalence, of Rep($A_5$).

torsion. The fusion rules for these quantum symmetries for trivial discrete torsion $\sigma$ are listed in Appendix A.

We remark that, these algebra objects can be alternatively obtained by starting from the $SU(2)_1$ theory where we have the $G = A_5$ symmetry, uplifting the algebra object (subgroup $H \subset A_5$ with a choice of discrete torsion) to the Drinfeld center $Z(G)$, and then project it back to $Rep(A_5)$ to get the corresponding algebra object. Such an analysis was explained in [PR24], where many examples was treated, including the $G = A_4$ case. We performed an analogous analysis for $G = A_5$, and we always find a result that is consistent with our subfactor analysis. In the future, it would be interesting to compare this Drinfeld center approach to the "polarization pair" approach of analyzing gaugings and SPTs introduced in [LYZ24].

## 6.2 Self-dualities of $SU(2)_1/A_5$ CFT under gauging Rep($A_5$)

From the last row of Table 5, we see that there exist gaugings of Rep($A_5$) whose dual quantum symmetry is Rep($A_5$) itself. This implies the possibility for a physical theory to be self-dual under gauging the algebra object $1 + X, 1 + X', 1 + X + X' + U + V$ and $1 + 2X + 2X' + 2U + 3V$.

An explicit setup is the exceptional $c = 1$ orbifold CFT $SU(2)_1/A_5$. As a $A_5$ orbifold theory, it naturally has a Rep($A_5$) symmetry. Similarly to what we have discussed in Section 5.2.2 for the Kosterlitz-Thouless CFT under gauging triality $\mathcal{C}^{\mathcal{T}}$, we can investigate gaugings of Rep($A_5$) as global manipulations on $SU(2)_1/A_5$ within the $c = 1$ conformal manifold.

| $H \subset A_5$ | Algebra object $A$ of $\mathrm{Rep}(A_5)$ | Quantum Symmetry $_A\mathcal{C}_A$ |
|---|---|---|
| $1$ | $1 + 3X + 3X' + 4U + 5V$ | $A_5$ |
| $\mathbb{Z}_2$ | $1 + X + X' + 2U + 3V$ | $\mathcal{C}(A_5, 1; \mathbb{Z}_2, 1)$ |
| $\mathbb{Z}_3$ | $1 + X + X' + 2U + V$ | $\mathcal{C}(A_5, 1; \mathbb{Z}_3, 1)$ |
| $(\mathbb{Z}_2)^2$ | $1 + U + 2V$ | $\mathcal{C}(A_5, 1; (\mathbb{Z}_2)^2, 1)$ |
| $(\mathbb{Z}_2^2)_{\mathrm{d.t.}}$ | $(1 + 3X + 3X' + 4U + 5V)_{\mathrm{d.t.}}$ | $\mathcal{C}(A_5, 1; (\mathbb{Z}_2)^2, \sigma)$ |
| $\mathbb{Z}_5$ | $1 + X + X' + V$ | $\mathcal{C}(A_5, 1; \mathbb{Z}_5, 1)$ |
| $S_3$ | $1 + U + V$ | $\mathcal{C}(A_5, 1; S_3, 1)$ |
| $D_5$ | $1 + V$ | $\mathcal{C}(A_5, 1; D_5, 1)$ |
| $A_4$ | $1 + U$ | $\mathcal{C}(A_5, 1; A_4, 1)$ |
| $(A_4)_{\mathrm{d.t.}}$ | $(1 + X + X' + 2U + V)_{\mathrm{d.t.}}$ | $\mathcal{C}(A_5, 1; A_4, \sigma)$ |
| $A_5$ | $1$ | $\mathrm{Rep}(A_5)$ |
| $(A_5)_{\mathrm{d.t.}}$ | $1 + X,\ 1 + X',\ 1 + X + X' + U + V,\ 1 + 2X + 2X' + 2U + 3V$ | $\mathrm{Rep}(A_5)$ |

Table 5: Gaugeable algebra objects of $\mathrm{Rep}(A_5)$, their associated $A_5$ gaugings, and dual fusion categories as quantum symmetries.

First, one investigates how gaugings of $A_5$ subgroups manipulate the $SU(2)_1$ theory, which is shown in Figure 22. We note that turning on discrete torsions for $\mathbb{Z}_2^2, A_4$ and $A_5$ do not lead to physically

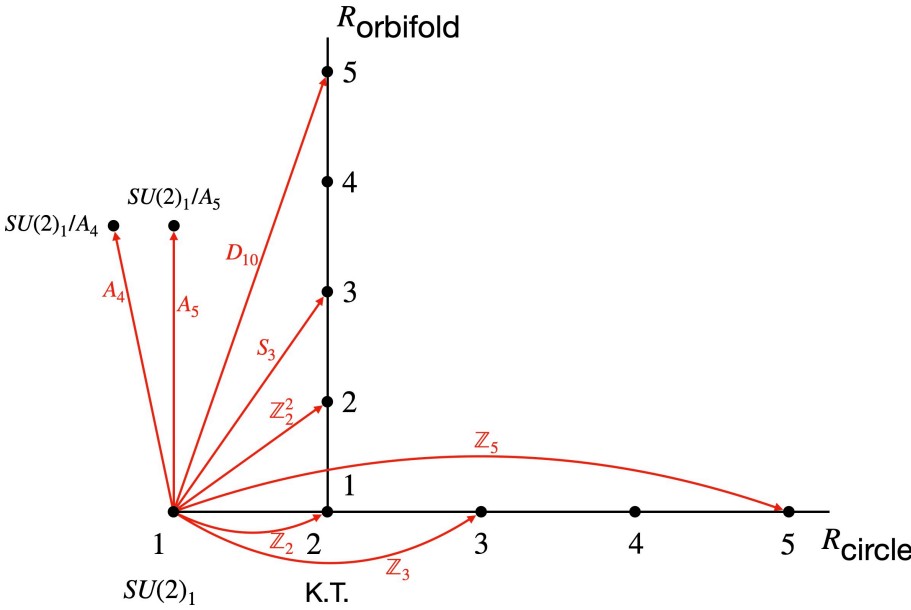

Figure 22: Global manipulations of $SU(2)_1$ via various gaugings of $A_5$.

distinct gaugings, due to the self-anomaly for the $SO(4)$ global symmetry of $SU(2)_1$ [TW24b]. Moreover, computing the fusion rule of the dual category $\mathcal{C}(A_5, 1; H, \sigma)$ for these three subgroups with non-trivial $\sigma$ will produce the same result as $\mathcal{C}(A_5, 1; H, 1)$[13]. Then, we can use the correspondence between gaugings of $A_5$ and those of $\mathrm{Rep}(A_5)$ in Table 5 to build global manipulations on $SU(2)_1/A_5$ theory, which is illustrated in Figure 23.

---

[13]More explicitly, the triple fusion of twisted $H - H$ modules can again be computed via [KMY97] using characters of the projective representations, and the discrete torsions can be determined by [Ost06] (see our 14). There, for the double coset containing id, one can always take $g = $ id and learn that $\psi^g$ is trivial. Then, only in $\mathcal{C}(A_5, 1; \mathbb{Z}_2 \times \mathbb{Z}_2, \sigma)$ there is a double-coset $HgH$ without identity where $H^2(H^g, U(1))$ is non-trivial. But here one can check that $[\psi^g] \in H^2(H^g, U(1))$ is trivial, following a similar computation as that in the case of $\mathcal{C}(A_4, 1; \mathbb{Z}_2 \times \mathbb{Z}_2, \sigma)$

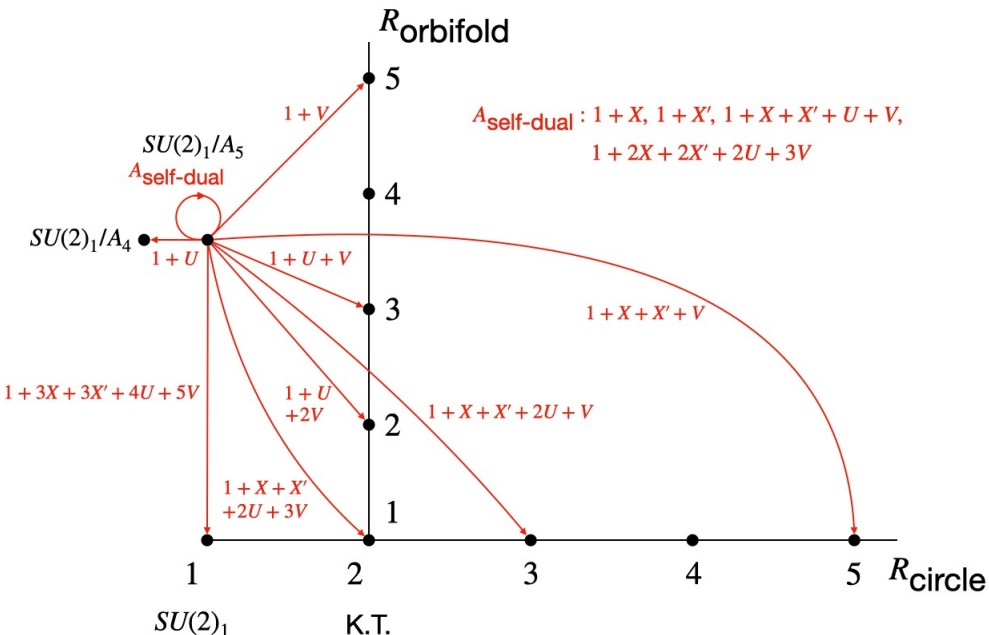

Figure 23: Global manipulations of $SU(2)_1/A_5$ on the $c = 1$ conformal manifold. Remark that there exist self-dual gaugings encoded by four Morita-equivalent algebra objects.

We can now see that $SU(2)_1/A_5$ is self-dual under gauging $A_{\text{self-dual}}$ in Figure 23, including four Morita-equivalent algebra objects. Under the half-space gauging (see, e.g., [LS21, CLS24, PLRS$^+$24a, DLWW24]), we can thus construct four duality defects $Y, Y', S$ and $T$, with self-fusion rules

$$
\begin{aligned}
Y \times Y &= 1 + X, \ Y' \times Y' = 1 + X', \\
S \times S &= 1 + X + X' + U + V, \\
T \times T &= 1 + 2X + 2X' + 2U + 3V.
\end{aligned}
\tag{66}
$$

The quantum dimensions of these new topological defects are $2, 2, 4$ and $6$, respectively. Recall that these are exactly dimensions for the four projective representations of $A_5$, which can be lifted to regular irreducible representations of $SL(2, 5)$. In fact, the resulting fusion ring consists of $1, X, X', U, V, Y, Y', S$ and $T$ simple objects is exactly that of a larger non-invertible symmetry $\text{Rep}(SL(2, 5))$. This can be understood from that $SL(2, 5)$ is a central extension of $A_5$

$$
1 \to \mathbb{Z}_2 \to SL(2, 5) \to A_5 \to 1.
\tag{67}
$$

The fusion category as representations of these groups then satisfy the following sequence

$$
1 \to \text{Rep}(A_5) \to \text{Rep}(SL(2, 5)) \to \text{Rep}(\mathbb{Z}_2) \to 1.
\tag{68}
$$

This is called *equivariantization* in the mathmatical terminology, see, e.g., [ENO11].

As discussed in [PLRS$^+$24b] (see also [DVVV89] for an earlier related discussion) about a similar phenomena in the context of extending Rep($A_4$) to Rep($SL(2,3)$)[14], the self-duality under gauging $A_{\text{self-dual}}$ implements a $\mathbb{Z}_2$-grading extension of Rep($A_5$) to Rep($SL(2,5)$).[15]

## 6.3  Classifying gapped phases and particle-soliton degeneracies

After discussing the Rep($A_5$) symmetry in CFTs, we now turn to the application of subfactors to gapped phases under Rep($A_5$) symmetry. These gapped phases are described by Rep($A_5$)-symmetric TQFTs, which are one-to-one corresponding to the gaugeable algebra objects in Table 5.

As explored in the previous examples, the vacuum structure of gapped phases is concluded in quiver diagrams, encoding representation theories for Rep($A_5$). Recall that the module categories are given by the odd parts in the principal graph. The resulting quiver diagrams are determined by the fusion of simple objects in Rep($A_5$) (even parts in the principal graph) with the even parts to which each simple object in $\mathcal{M}_A$ is connected.

Based on principal graphs in Figure 21, we obtain the quiver diagrams associated with all gaugeable algebra objects shown in Figure 24. We omit some quiver diagrams which can be easily obtained from relabeling quiver nodes. From the quiver diagrams, we conclude the vacuum structure of Rep($A_5$)-symmetric gapped phases and their particle-soliton degeneracies in Table 6.

| $A$ of Rep($A_5$) | Gapped phase | Particle-soliton degeneracy? |
|---|---|---|
| 1 | Rep($A_5$) SSB | yes |
| $1 + X$ | 4-vacuum SSB$_1$ | yes |
| $1 + U$ | 4-vacuum SSB$_2$ | yes |
| $1 + V$ | 4-vacuum SSB$_3$ | yes |
| $1 + U + 2V$ | 4-vacuum SSB$_4$ | no |
| $1 + X + X' + V$ | 5-vacuum SSB | yes |
| $1 + U + V$ | 3-vacuum SSB$_1$ | yes |
| $1 + X + X' + 2U + V$ | 3-vacuum SSB$_2$ | yes |
| $1 + X + X' + 2U + 3V$ | 2-vacuum SSB | yes |
| $1 + 3X + 3X' + 4U + 5V$ | Rep($A_5$) SPT | no |

Table 6: Classifications of Rep($A_5$)-symmetric gapped phases and their particle-soliton degeneracies.

# Acknowledgments

We thank Yuji Tachikawa for pointing us to helpful references, and for comments on an earlier version of the draft. We thank Yuan Miao, Daniel Robbins, and Xinping Yang for discussions on various

---

[14]See also [LSZ24] for a generalized extension known as $G$-alities.

[15]This Rep($SL(2,5)$) symmetry for $SU(2)_1/A_5$ theory can be embedded into the Verline lines associated with the chiral algebra $\mathcal{W}(2,36)$. This is a $\mathcal{W}$-algebra with 37 chiral primaries corresponding to 37 Verlinde lines. See [TW24b] for more details.

technical points. XY is supported by NSF grant PHY-2310588. HYZ is supported by WPI Initiative, MEXT, Japan at Kavli IPMU, the University of Tokyo. We thank the 2024 IHES Summer Workshop - Symmetries and Anomalies for the hospitality during which the project was initiated. We thank the 2024 Simons Physics Summer Workshop for the hospitality during the earlier stage of this project. HYZ thanks Simons Center for Geometry and Physics for hospitality during completion of the project.

# A    Fusion Rules for Dual Symmetries under $A_5$ Gauging

In this appendix, we give the fusion rule of group theoretic fusion categories of the form $\mathcal{C}(A_5, 1; H, 1)$, where $H$ is a proper subgroup of $A_5$. The result is explicitly computed by examining the triple fusion of $H - H$ bimodules using Sagemath, following [KMY97].

Non-trivial $\sigma$ in $\mathcal{C}(A_5, 1; H, \sigma)$ will arise for $H = \mathbb{Z}_2 \times \mathbb{Z}_2, A_4$ and $A_5$. But as explained in the main text, for all three cases we can explicitly check that, $\mathcal{C}(A_5, 1; H, \sigma)$ gives the same fusion rule as $\mathcal{C}(A_5, 1; H, 1)$ for all these cases.

Our notation and conventions will be explained in detail in the first example, but followed throughout this appendix.

## A.1    $\mathcal{C}(A_5, 1; \mathbb{Z}_2, 1)$

The dual fusion category in $\mathcal{C}(A_5, 1; \mathbb{Z}_2, 1)$ has four elements of quantum dimension 1 and 14 elements of quantum dimension 2. The four invertible elements form a group of $\mathbb{Z}_2 \times \mathbb{Z}_2$, which we label by $1, a, b, ab$. While the 14 on-invertible elements has fusion rules as in Table A.2. Our name of the 2-dimensional elements is such that, for example, $X_2^a$ and $X_2^{a'}$ are conjugate of each other, where as elements without a primed counterpart (such as $X_2^e$) are self-conjugate. In general, the subscript for $X$ always labels the dimension for a non-invertible element.

Here and in the rest of this appendix, we only give the fusion rule among a pair of non-invertible elements, from which one can obtain the fusion rules between one invertible element and one non-invertible element.

## A.2    $\mathcal{C}(A_5, 1; \mathbb{Z}_3, 1)$

For $\mathcal{C}(A_5, 1; \mathbb{Z}_3, 1)$, the 6 invertible elements form the group $S_3$, which we thus label by $1, a, a^2, b, ab, ab^2$. There are 6 non-invertible elements of 3 dimensions $(X_3^a, X_3^{a'}, X_3^b, X_3^c, X_3^d, X_3^e)$, whose fusion rules are given in Table 8.

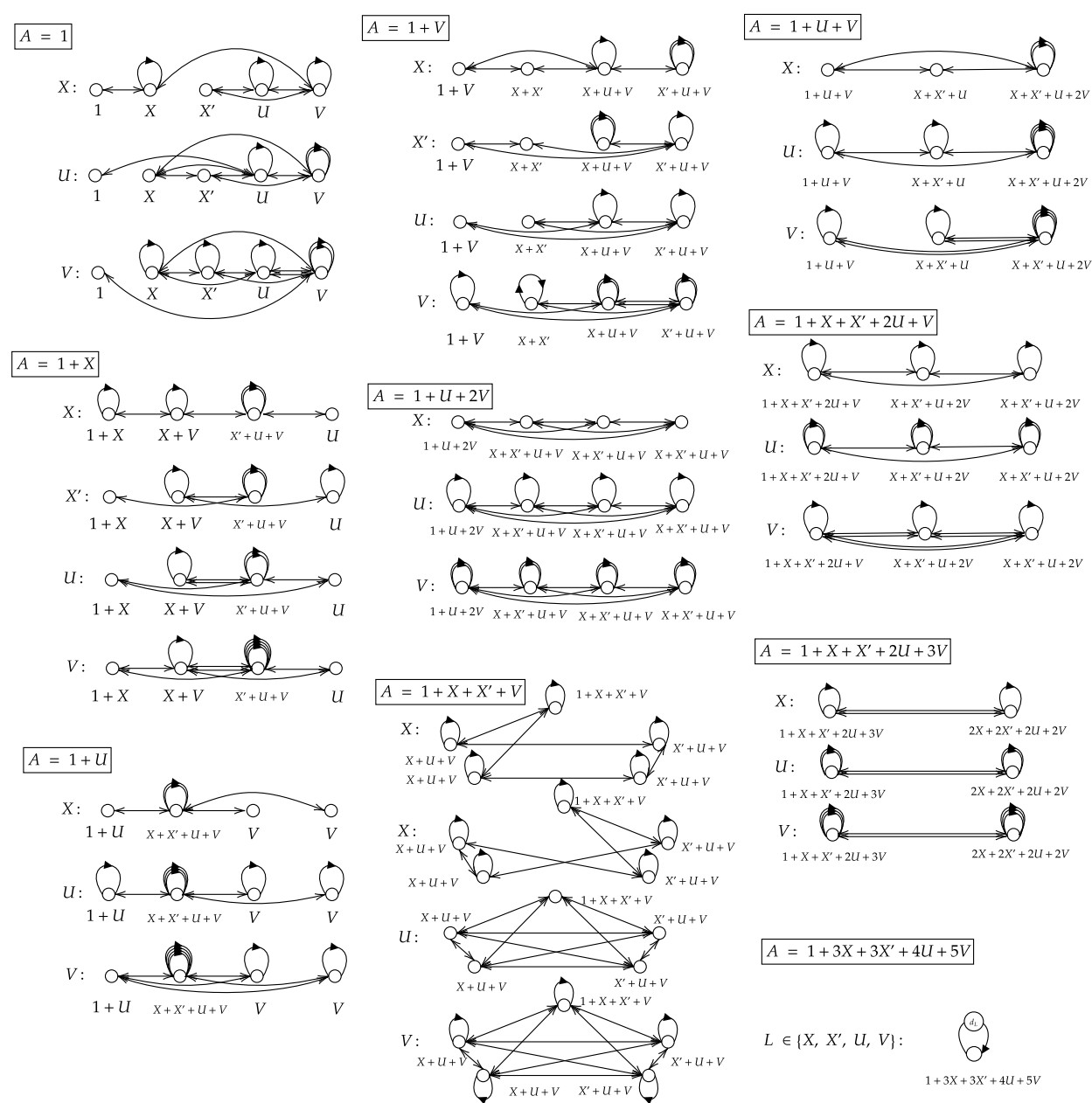

Figure 24: Quiver diagrams for representations of Rep($A_5$).

| | $X_2^a$ | $X_2^b$ | $X_2^e$ | $X_2^f$ | $X_2^c$ | $X_2^g$ | $X_2^h$ | $X_2^{b'}$ | $X_2^{c'}$ | $X_2^i$ | $X_2^{a'}$ | $X_2^j$ | $X_2^d$ | $X_2^{d'}$ |
|---|---|---|---|---|---|---|---|---|---|---|---|---|---|---|
| $X_2^a$ | $X_2^d+X_2^{a'}$ | $X_2^i+X_2^f$ | $X_2^b+X_2^g$ | $X_2^h+X_2^e$ | $X_2^j+X_2^b$ | $X_2^h+X_2^{c'}$ | $X_2^{b'}+ab+b$ | $X_2^c+X_2^g$ | $X_2^d+X_2^e$ | $X_2^{d'}+X_2^c$ | $1+X_2^i+a$ | $X_2^a+X_2^{b'}$ | $X_2^{d'}+X_2^j$ | $X_2^{c'}+X_2^{a'}$ |
| $X_2^b$ | $X_2^h+X_2^j$ | $X_2^{b'}+X_2^d$ | $X_2^a+X_2^g$ | $X_2^{a'}+X_2^b$ | $X_2^d+X_2^e$ | $X_2^i+X_2^c$ | $X_2^{d'}+X_2^{c'}$ | $1+X_2^h+a$ | $X_2^a+X_2^f$ | $ab+X_2^{a'}+b$ | $X_2^{c'}+X_2^g$ | $X_2^i+X_2^e$ | $X_2^{d'}+X_2^f$ | $X_2^{b'}+X_2^c$ |
| $X_2^e$ | $X_2^c+X_2^f$ | $X_2^{c'}+X_2^j$ | $1+a+X_2^e$ | $X_2^a+X_2^c$ | $X_2^a+X_2^f$ | $X_2^{b'}+X_2^{a'}$ | $X_2^{d'}+X_2^i$ | $X_2^{a'}+X_2^g$ | $X_2^j+X_2^b$ | $X_2^{d'}+X_2^h$ | $X_2^{b'}+X_2^g$ | $X_2^{c'}+X_2^b$ | $ab+X_2^d+b$ | $X_2^h+X_2^i$ |
| $X_2^f$ | $X_2^{b'}+ab+b$ | $X_2^{d'}+X_2^c$ | $X_2^{c'}+X_2^{a'}$ | $1+X_2^i+a$ | $X_2^j+X_2^b$ | $X_2^{d'}+X_2^j$ | $X_2^d+X_2^{a'}$ | $X_2^a+X_2^{b'}$ | $X_2^d+X_2^e$ | $X_2^i+X_2^f$ | $X_2^h+X_2^e$ | $X_2^c+X_2^g$ | $X_2^h+X_2^{c'}$ | $X_2^b+X_2^g$ |
| $X_2^c$ | $X_2^d+X_2^g$ | $X_2^h+X_2^a$ | $X_2^{b'}+X_2^j$ | $X_2^{d'}+X_2^e$ | $X_2^{c'}$x2 | $X_2^i+X_2^b$ | $X_2^d+X_2^g$ | $X_2^{a'}+X_2^f$ | $1+a+ab$ | $X_2^h+X_2^a$ | $X_2^{d'}+X_2^e$ | $X_2^{a'}+X_2^f$ | $X_2^i+X_2^b$ | $X_2^{b'}+X_2^j$ |
| $X_2^g$ | $X_2^e+X_2^b$ | $X_2^a+X_2^e$ | $X_2^a+X_2^b$ | $X_2^d+X_2^j$ | $X_2^h+X_2^{a'}$ | $1+a+X_2^g$ | $X_2^c+X_2^{a'}$ | $X_2^i+X_2^{c'}$ | $X_2^i+X_2^{b'}$ | $X_2^{b'}+X_2^{c'}$ | $X_2^h+X_2^c$ | $X_2^d+X_2^f$ | $X_2^j+X_2^f$ | $X_2^{d'}+ab+b$ |
| $X_2^h$ | $X_2^g+X_2^f$ | $X_2^{a'}+X_2^b$ | $X_2^i+X_2^d$ | $X_2^{d'}+X_2^a$ | $X_2^i+X_2^{b'}$ | $X_2^a+X_2^{c'}$ | $1+a+X_2^j$ | $X_2^d+X_2^c$ | $X_2^{d'}+X_2^g$ | $X_2^c+X_2^e$ | $ab+b+X_2^b$ | $X_2^h+X_2^j$ | $X_2^{b'}+X_2^e$ | $X_2^{c'}+X_2^f$ |
| $X_2^{b'}$ | $X_2^{c'}+X_2^e$ | $1+a+X_2^f$ | $X_2^c+X_2^j$ | $X_2^{c'}+X_2^d$ | $X_2^{d'}+X_2^g$ | $X_2^e+X_2^{a'}$ | $X_2^a+X_2^{b'}$ | $X_2^{d'}+X_2^b$ | $X_2^h+X_2^{a'}$ | $X_2^j+X_2^g$ | $X_2^i+X_2^f$ | $X_2^a+ab+b$ | $X_2^c+X_2^b$ | $X_2^h+X_2^d$ |
| $X_2^{c'}$ | $X_2^i+X_2^b$ | $X_2^d+X_2^g$ | $X_2^{a'}+X_2^f$ | $X_2^{b'}+X_2^j$ | $1+a+ab$ | $X_2^h+X_2^a$ | $X_2^i+X_2^b$ | $X_2^{d'}+X_2^e$ | $X_2^c$x2 | $X_2^d+X_2^g$ | $X_2^{b'}+X_2^j$ | $X_2^{d'}+X_2^e$ | $X_2^h+X_2^a$ | $X_2^{a'}+X_2^f$ |
| $X_2^i$ | $X_2^a+X_2^{b'}$ | $X_2^j+X_2^g$ | $X_2^h+X_2^d$ | $X_2^i+X_2^f$ | $X_2^{d'}+X_2^g$ | $X_2^c+X_2^b$ | $X_2^{c'}+X_2^e$ | $X_2^a+ab+b$ | $X_2^h+X_2^{a'}$ | $1+a+X_2^f$ | $X_2^{c'}+X_2^d$ | $X_2^{d'}+X_2^b$ | $X_2^e+X_2^{a'}$ | $X_2^c+X_2^j$ |
| $X_2^{a'}$ | $1+a+X_2^j$ | $X_2^c+X_2^e$ | $X_2^{c'}+X_2^f$ | $ab+b+X_2^b$ | $X_2^i+X_2^{b'}$ | $X_2^{b'}+X_2^e$ | $X_2^g+X_2^f$ | $X_2^h+X_2^j$ | $X_2^{d'}+X_2^g$ | $X_2^{a'}+X_2^b$ | $X_2^{d'}+X_2^a$ | $X_2^d+X_2^c$ | $X_2^a+X_2^{c'}$ | $X_2^i+X_2^d$ |
| $X_2^j$ | $X_2^{d'}+X_2^{c'}$ | $ab+X_2^{a'}+b$ | $X_2^{b'}+X_2^c$ | $X_2^{c'}+X_2^g$ | $X_2^d+X_2^e$ | $X_2^{d'}+X_2^f$ | $X_2^h+X_2^j$ | $X_2^i+X_2^e$ | $X_2^a+X_2^f$ | $X_2^{b'}+X_2^d$ | $X_2^{a'}+X_2^b$ | $1+X_2^h+a$ | $X_2^i+X_2^c$ | $X_2^a+X_2^g$ |
| $X_2^d$ | $X_2^{d'}+X_2^i$ | $X_2^{d'}+X_2^h$ | $X_2^h+X_2^i$ | $X_2^{b'}+X_2^g$ | $X_2^a+X_2^f$ | $ab+X_2^d+b$ | $X_2^c+X_2^f$ | $X_2^{c'}+X_2^b$ | $X_2^j+X_2^b$ | $X_2^{c'}+X_2^j$ | $X_2^a+X_2^c$ | $X_2^{a'}+X_2^g$ | $X_2^{b'}+X_2^{a'}$ | $1+a+X_2^e$ |
| $X_2^{d'}$ | $X_2^c+X_2^{a'}$ | $X_2^{b'}+X_2^{c'}$ | $X_2^{d'}+ab+b$ | $X_2^h+X_2^c$ | $X_2^h+X_2^{a'}$ | $X_2^j+X_2^f$ | $X_2^e+X_2^b$ | $X_2^d+X_2^f$ | $X_2^i+X_2^{b'}$ | $X_2^a+X_2^e$ | $X_2^d+X_2^j$ | $X_2^i+X_2^{c'}$ | $1+a+X_2^g$ | $X_2^a+X_2^b$ |

Table 7: Fusion rules of non-invertible elements in $\mathcal{C}(A_5, 1; \mathbb{Z}_2, 1)$.

| | $X_3^a$ | $X_3^{a'}$ | $X_3^b$ | $X_3^c$ | $X_3^d$ | $X_3^e$ |
|---|---|---|---|---|---|---|
| $X_3^a$ | $X_3^{a'}+X_3^c+X_3^e$ | $a^2+a+1+X_3^d\text{x2}$ | $ba^2+X_3^a\text{x2}+ba+b$ | $X_3^{a'}+X_3^c+X_3^b$ | $X_3^c+X_3^b+X_3^e$ | $X_3^{a'}+X_3^b+X_3^e$ |
| $X_3^{a'}$ | $a^2+X_3^b\text{x2}+a+1$ | $X_3^c+X_3^e+X_3^a$ | $X_3^c+X_3^e+X_3^d$ | $X_3^c+X_3^d+X_3^a$ | $ba^2+ba+X_3^{a'}\text{x2}+b$ | $X_3^d+X_3^e+X_3^a$ |
| $X_3^b$ | $X_3^c+X_3^e+X_3^d$ | $ba^2+ba+X_3^{a'}\text{x2}+b$ | $a^2+X_3^b\text{x2}+a+1$ | $X_3^d+X_3^e+X_3^a$ | $X_3^c+X_3^e+X_3^a$ | $X_3^c+X_3^d+X_3^a$ |
| $X_3^c$ | $X_3^{a'}+X_3^c+X_3^d$ | $X_3^c+X_3^b+X_3^a$ | $X_3^{a'}+X_3^d+X_3^e$ | $X_3^{a'}+a+X_3^a+a^2+1$ | $X_3^b+X_3^a+X_3^e$ | $ba^2+X_3^b+ba+b+X_3^d$ |
| $X_3^d$ | $ba^2+X_3^a\text{x2}+ba+b$ | $X_3^c+X_3^b+X_3^e$ | $X_3^{a'}+X_3^c+X_3^e$ | $X_3^{a'}+X_3^b+X_3^e$ | $a^2+a+1+X_3^d\text{x2}$ | $X_3^{a'}+X_3^c+X_3^b$ |
| $X_3^e$ | $X_3^{a'}+X_3^d+X_3^e$ | $X_3^b+X_3^a+X_3^e$ | $X_3^{a'}+X_3^c+X_3^d$ | $ba^2+X_3^b+ba+b+X_3^d$ | $X_3^c+X_3^b+X_3^a$ | $X_3^{a'}+a+X_3^a+a^2+1$ |

Table 8: Fusion rules of non-invertible elements in $\mathcal{C}(A_5,1;\mathbb{Z}_3,1)$.

## A.3 $\quad \mathcal{C}(A_5,1;\mathbb{Z}_2\times\mathbb{Z}_2,1)$

For $\mathcal{C}(A_5,1;\mathbb{Z}_2\times\mathbb{Z}_2,1)$, the 8 invertible elements form the group $D_4$, which we label by $1^0,1^x,1_{v,s,c}^0,1_{v,s,c}^x$. Here the superscript $0$ or $x$ indicate two double-cosets each containing 4 elements, and $(id),v,s,c$ indicates the representation of the centralizer which is isomorphic to $H=\mathbb{Z}_2\times\mathbb{Z}_2$. The three non-invertible elements of dimension 4 fuse according to the Table A.3.

|  | $X_4^a$ | $X_4^b$ | $X_4^c$ |
|---|---|---|---|
| $X_4^a$ | $1_s^0+1_v^0+X_4^c+X_4^b+X_4^a+1^0+1_c^0$ | $1^x+1_c^x+1_s^x+1_v^x+X_4^c+X_4^a+X_4^b$ | $1_c^y+1_s^y+1_v^y+X_4^c+X_4^a+1^y+X_4^b$ |
| $X_4^b$ | $1_c^y+1_s^y+1_v^y+X_4^c+X_4^a+1^y+X_4^b$ | $1_s^0+1_v^0+X_4^c+X_4^b+X_4^a+1^0+1_c^0$ | $1^x+1_c^x+1_s^x+1_v^x+X_4^c+X_4^a+X_4^b$ |
| $X_4^c$ | $1^x+1_c^x+1_s^x+1_v^x+X_4^c+X_4^a+X_4^b$ | $1_c^y+1_s^y+1_v^y+X_4^c+X_4^a+1^y+X_4^b$ | $1_s^0+1_v^0+X_4^c+X_4^b+X_4^a+1^0+1_c^0$ |

Table 9: Fusion rules of non-invertible elements in $\mathcal{C}(A_5,1;\mathbb{Z}_2\times\mathbb{Z}_2,1)$.

## A.4 $\quad \mathcal{C}(A_5,1;\mathbb{Z}_5,1)$

For $\mathcal{C}(A_5,1;\mathbb{Z}_5,1)$, we have five invertible elements forming the finite group of $D_5$ labelled by $1_{u^j}^0,1_{u^j}^x$, where $j=0,1,2,3,4$. The two non-invertible elements of dimension 5 fuse according to Table A.4.

|  | $X_5^a$ | $X_5^b$ |
|---|---|---|
| $X_5^a$ | $1_{u^2}^0+X_5^a\text{x}2+1_{u^4}^0+1_u^0+1_{u^3}^0+X_5^b\text{x}2+1^0$ | $X_5^a\text{x}2+1_{u^2}^x+1^x+1_{u^4}^x+1_u^x+1_{u^3}^x+X_5^b\text{x}2$ |
| $X_5^b$ | $X_5^a\text{x}2+1_{u^2}^x+1^x+1_{u^4}^x+1_u^x+1_{u^3}^x+X_5^b\text{x}2$ | $1_{u^2}^0+X_5^a\text{x}2+1_{u^4}^0+1_u^0+1_{u^3}^0+X_5^b\text{x}2+1^0$ |

Table 10: Fusion rules of non-invertible elements in $\mathcal{C}(A_5,1;\mathbb{Z}_5,1)$.

## A.5 $\quad \mathcal{C}(A_5,1;S_3,1)$

For $\mathcal{C}(A_5,1;S_3,1)$, we have two invertible elements labelled by $1,1_-$ forming $\mathbb{Z}_2$. The remaining non-invertible elements of dimension $2,3,3,6$ has fusion rules that can be described in the following table A.5.

|  | $X_2$ | $X_6$ | $X_3^+$ | $X_3^-$ |
|---|---|---|---|---|
| $X_2$ | $X_2+1_-+1$ | $X_6\text{x}2$ | $X_3^-+X_3^+$ | $X_3^-+X_3^+$ |
| $X_6$ | $X_6\text{x}2$ | $X_3^-\text{x}2+1_-+X_2\text{x}2+X_3^+\text{x}2+X_6\text{x}3+1$ | $X_3^-+X_3^++X_6\text{x}2$ | $X_3^-+X_3^++X_6\text{x}2$ |
| $X_3^+$ | $X_3^-+X_3^+$ | $X_3^-+X_3^++X_6\text{x}2$ | $X_2+X_6+1$ | $X_2+1_-+X_6$ |
| $X_3^-$ | $X_3^-+X_3^+$ | $X_3^-+X_3^++X_6\text{x}2$ | $X_2+1_-+X_6$ | $X_2+X_6+1$ |

Table 11: Fusion rules of non-invertible elements in $\mathcal{C}(A_5,1;S_3,1)$.

## A.6 $\mathcal{C}(A_5, 1; D_5, 1)$

For $\mathcal{C}(A_5, 1; D_5, 1)$, we have two invertible elements $1, 1_-$ forming the group $\mathbb{Z}_2$. The non-invertible elements of dimension $2, 2, 5, 5$ fuse according to table A.6.

| | $X_2^s$ | $X_2^c$ | $X_5^+$ | $X_5^-$ |
|---|---|---|---|---|
| $X_2^s$ | $1 + X_2^c + 1_-$ | $X_2^s + X_2^c$ | $X_5^- + X_5^+$ | $X_5^- + X_5^+$ |
| $X_2^c$ | $X_2^s + X_2^c$ | $1 + X_2^s + 1_-$ | $X_5^- + X_5^+$ | $X_5^- + X_5^+$ |
| $X_5^+$ | $X_5^- + X_5^+$ | $X_5^- + X_5^+$ | $1 + X_2^s + X_5^- \text{x2} + X_5^+ \text{x2} + X_2^c$ | $X_2^s + X_5^- \text{x2} + 1_- + X_5^+ \text{x2} + X_2^c$ |
| $X_5^-$ | $X_5^- + X_5^+$ | $X_5^- + X_5^+$ | $X_2^s + X_5^- \text{x2} + 1_- + X_5^+ \text{x2} + X_2^c$ | $1 + X_2^s + X_5^- \text{x2} + X_5^+ \text{x2} + X_2^c$ |

Table 12: Fusion rules of non-invertible elements in $\mathcal{C}(A_5, 1; D_5, 1)$.

## A.7 $\mathcal{C}(A_5, 1; A_4, 1)$

For $\mathcal{C}(A_5, 1; A_4, 1)$, we have we have three invertible elements $1, a, a^2$ forming $\mathbb{Z}_3$. The non-invertible elements of dimension $3, 4, 4, 4$ fuse according to table A.7.

| | $X_3$ | $X_4^1$ | $X_4^u$ | $X_4^{u^2}$ |
|---|---|---|---|---|
| $X_3$ | $1 + a + a^2 + X_3 \text{x2}$ | $X_4^1 + X_4^{u^2} + X_4^u$ | $X_4^1 + X_4^{u^2} + X_4^u$ | $X_4^1 + X_4^{u^2} + X_4^u$ |
| $X_4^1$ | $X_4^1 + X_4^{u^2} + X_4^u$ | $1 + X_4^1 + X_4^u + X_3 + X_4^{u^2}$ | $X_4^1 + X_4^u + X_4^{u^2} + X_3 + a$ | $X_4^1 + X_4^u + a^2 + X_3 + X_4^{u^2}$ |
| $X_4^u$ | $X_4^1 + X_4^{u^2} + X_4^u$ | $X_4^1 + X_4^u + a^2 + X_3 + X_4^{u^2}$ | $1 + X_4^1 + X_4^u + X_3 + X_4^{u^2}$ | $X_4^1 + X_4^u + X_4^{u^2} + X_3 + a$ |
| $X_4^{u^2}$ | $X_4^1 + X_4^{u^2} + X_4^u$ | $X_4^1 + X_4^u + X_4^{u^2} + X_3 + a$ | $X_4^1 + X_4^u + a^2 + X_3 + X_4^{u^2}$ | $1 + X_4^1 + X_4^u + X_3 + X_4^{u^2}$ |

Table 13: Fusion rules of non-invertible elements in $\mathcal{C}(A_5, 1; A_4, 1)$.

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
