# Peer review of "von Neumann Subfactors and Non-invertible Symmetries"

_SciPost Physics_

## Round 2 · Referee Report · Anonymous (Referee 1) · 2025-7-13

Strengths

1- The authors give explicit examples of gauging non-invertible symmetries in CFTs using the method developed in subfactor theory.

2- The authors give several physical interpretations of the principal graph and quiver diagram and connect them to the generalized gauging, particle-soliton degeneracy, and self-duality.

Weaknesses

1- Lack of presentation clarity: (a) Notations and concepts are not defined or are not consistent throughout the paper. For instance, the central object in the paper is the fusion matrix F, but \xi_i,j are not defined around eq.9. After section 4, the authors introduce the reduced fusion matrix F^r_A; what is the relationship between them? Another confusing thing is that there are various products in section 2. How are they defined? (b) It is also unclear which part is the review and which part is the new results due to the authors; it would be better to state them clearly.

2- The authors advertise using the subfactor theory to study the CFT. I think there are still ongoing big questions: can one construct a CFT from a subfactor, or are all the subfactors from CFTs? Since the examples in the current paper are all group-theoretical fusion categories, I'm wondering if there is a concrete way to construct the CFT from these subfactors with group-theoretical data.

3- The gaugeable algebra and module categories of the group theoretical fusion category are known and classified in the literature. I'm wondering if there's more we can get from the subfactor method. Before that, (a) how can one tell two algebraic objects are Morita equivalent from the subfactor method? (b) How can one get 3 maximal gauging in RepD8 from the subfactor method, it seems only one in the current presentation. (c) How does this method connect to the NIM-reps method?

4- Can one extract whether the fusion category admits z2 or other group extension or not from the principal graph? Relatedly, how does the extension data manifest, namely, the fractionalization class and the FS indicator? Is there an argument for why the self-duality of SU(2)_1/A_5 CFT admits the trivial fractionalization class and trivial FS indicator?

Report

The authors use methods developed in the subfactor theory to study the generalized gauging of non-invertible symmetry in the conformal field theories. In particular, the authors use the principal graph, relatedly the fusion matrix, to find the gaugeable algebraic objects. They use the principal graph to derive a quiver diagram and discuss the symmetry actions and particle-soliton degeneracies. The questions are raised in the Weaknesses section.

Requested changes

Please see the Weaknesses section.

Recommendation

Ask for major revision

  • validity: good
  • significance: good
  • originality: ok
  • clarity: low
  • formatting: reasonable
  • grammar: perfect

Author:  Xingyang Yu  on 2025-10-29  [id 5963]

(in reply to Report 1 on 2025-07-13)

Report I:

1- Lack of presentation clarity: (a) Notations and concepts are not defined or are not consistent throughout the paper. For instance, the central object in the paper is the fusion matrix F, but \xi_i,j are not defined around eq.9. After section 4, the authors introduce the reduced fusion matrix F^r_A; what is the relationship between them? Another confusing thing is that there are various products in section 2. How are they defined? (b) It is also unclear which part is the review and which part is the new results due to the authors; it would be better to state them clearly.

(a) We added contents around Eq.(9) to explain the definition of fusion matrix $F$ in terms of the basis elements $\xi_{i,j}$. We also added context in Section 4 (between equation (27) and (28)) to explain the relation between the reduced fusion matrix $F^r$ and the (unreduced) fusion matrix $F$.

(b) We also added sentences (at the beginning of section 2.1 and 2.2) clarifying the entire section 2 covers standard contents of von Neumann algebras and subfactors.

2- The authors advertise using the subfactor theory to study the CFT. I think there are still ongoing big questions: can one construct a CFT from a subfactor, or are all the subfactors from CFTs? Since the examples in the current paper are all group-theoretical fusion categories, I'm wondering if there is a concrete way to construct the CFT from these subfactors with group-theoretical data.

We thank the referee for raising this interesting question. Intuitively speaking, at least for chiral CFTs captured by vertex operator algebras, they are widely believed to be equivalently described by the local conformal nets, which in turn admit a subfactor description. We briefly comment on the possibility of building CFTs from tensor categories at the end of section 3, and refer to the literature in the case of the group-theoretical data.

3- The gaugeable algebra and module categories of the group theoretical fusion category are known and classified in the literature. I'm wondering if there's more we can get from the subfactor method. Before that, (a) how can one tell two algebraic objects are Morita equivalent from the subfactor method? (b) How can one get 3 maximal gauging in RepD8 from the subfactor method, it seems only one in the current presentation. (c) How does this method connect to the NIM-reps method?

(a) One necessary condition for two algebra objects being Morita equivalent is they share the same quiver diagram extracted from the subfactor principal graphs. This translates to the fact that two Morita equivalent algebra objects, by definition, correspond to the same module category.

(b) From the principal graphs, one is not able to distinguish three maximal gaugings of Rep(D8). However, Q-system of a given subfactor contains more information than the principal graph. We have added a paragraph above Table 2 to clarify this limitation and leave the application of the full Q-system to gauging non-invertible symmetries to future study.

(c) In our language, the NIM-representation of the fusion category is completely captured by the quiver diagrams that we introduced in Section 2.3 (see figure 5 and many other examples in Section 4), whose derivation made use of the principal diagram. So these two objects are indeed closely connected.

4- Can one extract whether the fusion category admits z2 or other group extension or not from the principal graph? Relatedly, how does the extension data manifest, namely, the fractionalization class and the FS indicator? Is there an argument for why the self-duality of SU(2)_1/A_5 CFT admits the trivial fractionalization class and trivial FS indicator?

So far we do not have a good strategy to investigate the extension and its fracionalization class of a fusion category using the subfactor language. In Section 6, we indeed do not claim the extended fusion category to be Rep(SL(2,5)), but only that the fusion category has the fusion ring of Rep(SL(2,5)). We have added a paragraph to further clarify this point and mentioned as an interesting future direction of studying extension classes of fusion categories from the subfactor perspective.

---

## Round 2 · Referee Report · Anonymous (Referee 2) · 2025-7-24

Report

The paper discusses various applications of the subfactor theory in the study of 1+1d conformal and topological field theories. In particular, the authors utilize the known mathematical connection between the theory of fusion categories and that of subfactors to classify Frobenius algebra objects and module categories of various fusion categories. Physically, Frobenius algebra objects correspond to different ways of gauging a (non-invertible) global symmetry, and module categories classify different symmetry breaking patterns.

Although the relation between the subfactor theory and fusion categories is well-known in mathematics, in the physics literature this is not very much explored. It seems to be a valuable contribution by the authors to emphasize the utility of this mathematical connection in carrying out certain physically relevant computations in the study of (generalized) global symmetries of 1+1d systems. Therefore, the referee believes that the paper deserves to be published in SciPost, once the comments below are addressed.

Requested changes

- Sections 2.2 and 2.3 do not discuss original results by the authors, but are mostly based on Ref. [BKLR15]. The general presentation in these sections is not very comprehensible, and discussions are hard to follow without directly consulting Ref. [BKLR15]. It would be great if these sections can be revised for improved clarity.
- At the beginning of Section 2.2, it will be helpful to define what are DHR homomorphisms (or endomorphisms), and then explain why they form a fusion category. Without this, some of the following discussions seem to be hard to follow.
- The discussion that starts from the last paragraph on page 7 refers to certain pictures in Ref. [BKLR15], which is somewhat inconvenient since the discussion is hard to follow unless the reader opens Ref. [BKLR15] and looks for the corresponding pictures. Perhaps this can be improved, by adding similar and/or additional figures in the manuscript.
- On page 9, $\xi_{i,j}$ and the inner product $( \cdot , \cdot )$ are not defined.
- On page 15, the authors say, “… the practical computation … is very challenging in general, and one needs other mathematical tools to even do this computation.” Although it is true that in practice it is challenging to find a module category of a given fusion category, it does not seem true that this requires any additional mathematical tools. In principle, module categories can be found by solving the (module) pentagon equations, which are just algebraic equations.
- On page 16, there is a sentence “… there is a one-to-one correspondence between its gaugeable algebra objects A to C-symmetric TQFTs.” To be more precise (as the authors are of course aware), the one-to-one correspondence is between Morita equivalence classes of algebra objects and C-symmetric TQFTs.
- In Eq. (3), it appears that $1_\theta$ is not defined.
- Above Eq. (6), instead of “identity homomorphsim” maybe “injective homomorphism” is more appropriate?

Recommendation

Ask for minor revision

  • validity: -
  • significance: -
  • originality: -
  • clarity: -
  • formatting: -
  • grammar: -

Author:  Xingyang Yu  on 2025-10-29  [id 5964]

(in reply to Report 2 on 2025-07-24)

  • Sections 2.2 and 2.3 do not discuss original results by the authors, but are mostly based on Ref. [BKLR15]. The general presentation in these sections is not very comprehensible, and discussions are hard to follow without directly consulting Ref. [BKLR15]. It would be great if these sections can be revised for improved clarity.

We modified the discussion in Section 2.2 and 2.3 by giving more detailed explanations and including some pictures from [BKLR15], which is hopefully more accessible for the readers.

  • At the beginning of Section 2.2, it will be helpful to define what are DHR homomorphisms (or endomorphisms), and then explain why they form a fusion category. Without this, some of the following discussions seem to be hard to follow.

We now briefly reivew DHR endomorphisms and how it associates to fusion categories at the beginning of Section 2.2.

  • The discussion that starts from the last paragraph on page 7 refers to certain pictures in Ref. [BKLR15], which is somewhat inconvenient since the discussion is hard to follow unless the reader opens Ref. [BKLR15] and looks for the corresponding pictures. Perhaps this can be improved, by adding similar and/or additional figures in the manuscript.

We have included new Figures 1, 2, and 3 for illustration.

  • On page 9, $\xi_{i,j}$ and the inner product $(\cdot, \cdot)$ are not defined.

We have defined the inner product via Eq.(9).

  • On page 15, the authors say, “… the practical computation … is very challenging in general, and one needs other mathematical tools to even do this computation.” Although it is true that in practice it is challenging to find a module category of a given fusion category, it does not seem true that this requires any additional mathematical tools. In principle, module categories can be found by solving the (module) pentagon equations, which are just algebraic equations.

We have edited that sentence to make it clear that we are applying subfactors as an alternative method to quickly get necessary conditions for module categories.

  • On page 16, there is a sentence “… there is a one-to-one correspondence between its gaugeable algebra objects A to C-symmetric TQFTs.” To be more precise (as the authors are of course aware), the one-to-one correspondence is between Morita equivalence classes of algebra objects and C-symmetric TQFTs.

We changed our statement accordingly.

  • In Eq. (3), it appears that $1_\theta$ is not defined.

We added the definition of $1_\theta$ as the identify morphism after equations (2)-(4).

  • Above Eq. (6), instead of “identity homomorphsim” maybe “injective homomorphism” is more appropriate?

We have changed the name $\iota$ to "injective homomorphism" as suggested - indeed, the term "injective homomorphism" is more suitable for $1_\theta$.

---

## Editorial Decision

resubmitted